## RESEARCH ARTICLE

# Functional requirements of the liver isoform of phosphofructokinase-1 in breast cancer cell migration

Heather L. Hansen and Bradley A. Webb*

## ABSTRACT

Increased aerobic glycolysis and increased cell motility are hallmarks of metastatic cancer. Migrating cancer cells are highly polarized, suggesting that glycolytic enzymes could be spatially regulated. Here, we investigated the role of the liver isoform of the 'gatekeeper' glycolytic enzyme phosphofructokinase-1 (PFKL) in breast cancer cell migration. Depletion of PFKL significantly decreased migration velocity and directional sensing. We have observed the localization of PFKL to lamellipodia of migrating breast cancer cells, where it colocalized with hexokinase-2 and pyruvate kinase M2. We then investigated the functional requirements of PFKL for directional migration. First, we found that expression of catalytically inactive PFKL or indirect pharmacological inhibition of PFKL activity significantly decreased directional migration. Second, we discovered that disrupting PFKL filament formation by expression of a filament-incompetent mutant decreased PFKL recruitment to lamellipodia and directional sensing, without altering migration velocity. These findings indicate that both catalytic activity and subcellular localization are required for directional migration in breast cancer cells. These results suggest a novel function of PFKL filaments in cells and provide insight into the function of compartmentalized glycolysis in the cytoplasm.

KEY WORDS: Phosphofructokinase-1, PFKL, Cell migration, Chemotaxis, Glycolysis, Cancer

## INTRODUCTION

The metastasis of cancer cells to distal sites is a major contributor to the progression and mortality of cancer. Metastasis is a multi-step process by which cancer cells detach from a primary tumor, invade the surrounding tissue and disseminate to distant organs (Martin et al., 2013). A key step of the metastatic process is chemotaxis, the directed migration of cancer cells in response to chemoattractants such as growth factors. Chemotactic signaling guides tumors toward blood vessels for intravasation and aids in their exit into distant tissues during extravasation (Roussos et al., 2011; Stuelten et al., 2018).

Cell migration is energetically demanding, requiring ATP-dependent processes such as actin cytoskeletal remodeling, myosin contraction, focal adhesion turnover and ion transport (DeWane et al., 2021; Pourjafar and Tiwari, 2024; Santos et al., 2023; Zhang et al., 2022a). The altered metabolic program of cancer cells, known as the Warburg effect, is characterized by increased glycolysis even in the presence of oxygen, and generation of ATP to support the increased proliferation and migration of cancer cells (Lunt and Heiden, 2011; Mikawa et al., 2015; Shiraishi et al., 2014). Increased glycolysis is associated with increased rates of cytoskeletal remodeling, greater traction forces and greater migration of cancer cells (Shiraishi et al., 2014). Further, previous studies on tumor cell metabolism suggest that increased glycolysis contributes to cancer cell chemotaxis (Beckner et al., 1990). In motile cells, glycolysis is proposed to be localized adjacent to the plasma membrane to locally generate ATP (DeWane et al., 2021; Ozawa et al., 2015; Shiraishi et al., 2014; Zhan et al., 2025). Enzymes of the glycolytic pathway have been observed at leading-edge lamellipodia and dorsal ruffles, where they are thought to locally generate ATP to fuel actin remodeling, membrane ruffling and focal adhesion turnover (De Bock et al., 2013; Zhan et al., 2025). Assemblies of glycolytic enzymes have been reported in multiple epithelial cell types and appear to form in response to a variety of stimuli, including epidermal growth factor (EGF) signaling, cell cycle progression, hypoxia and serum starvation (Jeon et al., 2023, 2022; Jin et al., 2017; Lynch et al., 2024). It is suggested that these assemblies modulate glycolytic flux in response to the energetic needs of the cell. Together, these data support the concept that glycolysis is not uniformly distributed in the cytoplasm, but rather compartmentalized to meet local energy demands during migration.

Glycolytic flux is controlled by highly regulated rate-limiting enzymes, one of the most important being phosphofructokinase-1 (PFK1). PFK1, referred to as the 'gatekeeper' of glycolysis, consumes one ATP to convert fructose-6-phosphate (F6P) to fructose-1,6-bisphosphate (F-1,6-bP). PFK1 is highly regulated through multiple mechanisms, including allosteric regulation by small molecules, post-translational modifications, and transcriptional and post-translational modulation of protein levels (Feng et al., 2020; Schöneberg et al., 2013; Yang et al., 2019; Yi et al., 2012). There are three human isoforms of PFK1: liver (PFKL), muscle (PFKM), and platelet (PFKP). The isoforms share ∼70% sequence identity, with this 30% difference hypothesized to impart isoform-specific activity, regulation and non-enzymatic functions (Fernandes et al., 2020). A unique property of PFKL is the ability to form elongated filaments *in vitro* (Webb et al., 2017). In cells, PFKL filaments appear to regulate the assembly of PFKL into biomolecular condensates. To study how PFKL filaments alter the subcellular localization of PFKL in cells, we generated a filament-incompetent mutant and found that it does not form punctate assemblies in the cytoplasm like wild-type PFKL (Lynch et al., 2024). While we show that filaments are critical for organizing PFKL within the cytoplasm, their function remains incompletely understood.

In this study, we investigate the requirement of PFKL in directional migration of MDA-MB-231 triple-negative human breast cancer cells to better understand how PFKL localization within the cytoplasm supports its function. We show that PFKL expression and catalytic activity are required for efficient directional migration. Further, we

Department of Biochemistry, West Virginia University School of Medicine, Morgantown, WV 26506, USA.

*Author for correspondence (bradley.webb@hsc.wvu.edu)

H.L.H., 0000-0002-5254-4662; B.A.W., 0000-0002-5299-2852

show that the ability to form filaments is necessary for PFKL localization to lamellipodia, and that this localization is required for directional sensing.

## RESULTS
### PFKL is required for directional cell migration
Glycolysis is a key source of ATP for energetically demanding processes that support cell migration such as cytoskeletal remodeling and the formation of membrane protrusions (De Bock et al., 2013; DeWane et al., 2021; Santos et al., 2023). However, little is known about the contribution of individual PFK1 isoforms. We have previously observed that PFKL–EGFP is recruited to distinct membrane domains in MTLn3 rat adenocarcinoma cells, and we hypothesize that PFKL plays a key role in directional cell migration (Webb et al., 2017). Here, we asked whether PFKL is required for chemotaxis in the human breast cancer cell line MDA-MB-231. This cell line is highly migratory and has been shown to rely heavily on upregulated glycolysis to sustain increased cell proliferation and migration (Price, 1996; Ye et al., 2013).

We sought to determine whether PFKL is required for cell migration by depleting PFKL using RNA interference (RNAi). MDA-MB-231 cells were transfected with control short interfering RNA (siRNA) or siRNA targeting PFKL, and the level of PFKL expression was determined by western blot analysis. Control siRNA (siCTRL)-transfected cells showed no difference in PFKL levels compared to non-transfected cells (Fig. 1A). Cells transfected with PFKL siRNA (siPFKL) showed a greater than 90% decrease in PFKL protein levels between 48 h and 96 h after transfection. We observed an increase in overall PFKL expression in control cells at 72 h and 96 h after transfection. This increase might reflect higher confluency, as metabolic enzyme expression, including that of PFKL, has been linked to mechanotransduction pathways responsive to changes in cellular stiffness (Park et al., 2020). Depletion of PFKL did not cause a compensatory increase in the expression of other PFK1 isoforms (Fig. S1A). To confirm the loss of PFKL, we performed a PFK1 activity assay to measure the total PFK1 activity in cell lysates. siCTRL-transfected cells showed no difference in F-1,6-bP production compared to non-transfected cells (Fig. S1B). Lysates from cells transfected with siPFKL showed ~50% PFK1 activity levels compared to those of control cell lysates, consistent with the western blot assays (Fig. S1B). We next determined how depletion of PFKL impacted glucose flux. First, we measured glycolytic flux as estimated by lactic acid excretion and found no significant difference between the cell types (Fig. S1B). To confirm these results, the cellular rate of glycolysis was determined by the extracellular acidification rate (ECAR) and was monitored in real time using Seahorse XF assays (Fig. 1B; Fig. S1C). Interestingly, no differences in glycolysis, glycolytic capacity or glycolytic reserve were observed between the treatments. This suggests compensation for the loss of PFKL through allosteric activation of the remaining PFK1 present, or that cells are producing lactate or protons through alternative pathways (Schmidt et al., 2021). The ratio of mitochondrial activity (oxygen consumption rate, OCR) to glycolytic activity (ECAR) was slightly elevated in siPFKL-transfected cells, consistent with an increased reliance on mitochondrial metabolism to compensate for loss of PFK1 activity in the cell (Fig. 1C).

As biosynthetic precursors generated from glycolysis are required for cell proliferation, we asked whether depletion of PFKL altered proliferation. We monitored cell number for 96 h after siRNA transfection. Cells transfected with siCTRL showed no difference in cell number compared to non-transfected controls (Fig. S1D,E). Cultures transfected with siPFKL had significantly fewer cells at

72 h and 96 h after transfection. Together, these data show that, despite the minor impact on estimated glycolytic flux, depletion of PFKL caused a significant decrease in cell proliferation.

To determine how depletion of PFKL impacted cell migration, we performed a single-cell chemotaxis assay to mitigate the effect of PFKL depletion on proliferation (Zengel et al., 2011). Additionally, experiments were started at 48 h after transfection, at which point protein levels of PFKL were significantly reduced but no significant difference in proliferation was observed. We measured the ability of cells plated in low-serum medium to migrate up an EGF gradient, with the hypothesis that PFKL depletion will cause a decrease in chemotaxis. Using a spider diagram to visualize the tracks of individual cells, we found that non-transfected control cells predominantly showed net migration up the EGF gradient (black tracks; 94.2% of cells) with only a few cells showing net migration down the gradient (red tracks) (Fig. 1D). Transfection with siCTRL had little impact on chemotaxis, with 88.5% of cells showing migration up the EGF gradient. In contrast, siRNA depletion of PFKL ablated the ability of cells to sense the EGF gradient, with an equal number of cells migrating up or down the EGF gradient, indicating that the cells were randomly migrating rather than directionally migrating towards the chemoattractant source. To further quantify the impact on directional sensing, we determined the forward migration index parallel (X-FMI) and perpendicular (Y-FMI) to the applied EGF gradient. In our experiments, the closer X-FMI is to −1, the stronger the directional migration up the EGF gradient, whereas values closer to 0 indicate weaker directional migration. As a control, the Y-FMI should be close to 0 as there is no perpendicular EGF gradient. Non-transfected and siCTRL-transfected cells showed a strong migration up the EGF gradient, with an X-FMI of −0.40±0.02 and −0.39±0.02 (mean±s.e.m.), respectively. In contrast, PFKL-depleted cells showed a significant loss of directional sensing, with an X-FMI of −0.03±0.03. No cell line showed a preference for migration perpendicular to the gradient, with a Y-FMI of near 0 for all treatments. Finally, PFKL-depleted cells also showed a decrease in migration velocity compared to control cells: non-transfected and siCTRL-transfected cells migrated at a 0.61±0.22 and 0.52±0.21 µm/min (mean±s.e.m.), respectively, whereas siPFKL-transfected cells migrated at 0.39±0.15 µm/min (Fig. 1E; Table S1). Together, these data demonstrate that PFKL is required for chemotaxis and that a loss of PFKL causes a decrease in both rate of migration and the ability to migrate up the EGF gradient.

We confirmed the effects of the siRNA knockdown using short hairpin RNA (shRNA) targeting human PFKL. MDA-MB-231 cells were transduced with either a non-targeting control shRNA (shScramble) or PFKL shRNA (shPFKL), and a pool of antibiotic-resistant cells was generated. Cells transduced with shPFKL had a significant decrease in PFKL levels compared to shScramble cells (Fig. S1F; 83.9%±6.9% depletion, mean±s.e.m.; P=0.01). There was no increase in the levels of PFKM or PFKP in response to PFKL depletion (Fig. S1G). We next quantified the impact of stable PFKL knockdown on proliferation (Fig. S1H). In contrast to siRNA knockdown assays, no significant difference in cell proliferation was observed in shPFKL cells when compared to parental or shScramble controls. The cause of this discrepancy is not known but might be caused by off-target effects of the PFKL siRNA or long-term adaptation of the cell lines; this is an area of future inquiry. Finally, we asked whether chemotaxis was altered in shPFKL-transduced cells. In contrast to parental and shScramble cell lines, PFKL shRNA-expressing cells had a significant decrease in the ability to migrate up an EGF gradient. Spider diagrams show 95.0% of parental cells and 89.3% of shScramble cells migrating up the EGF gradient, compared

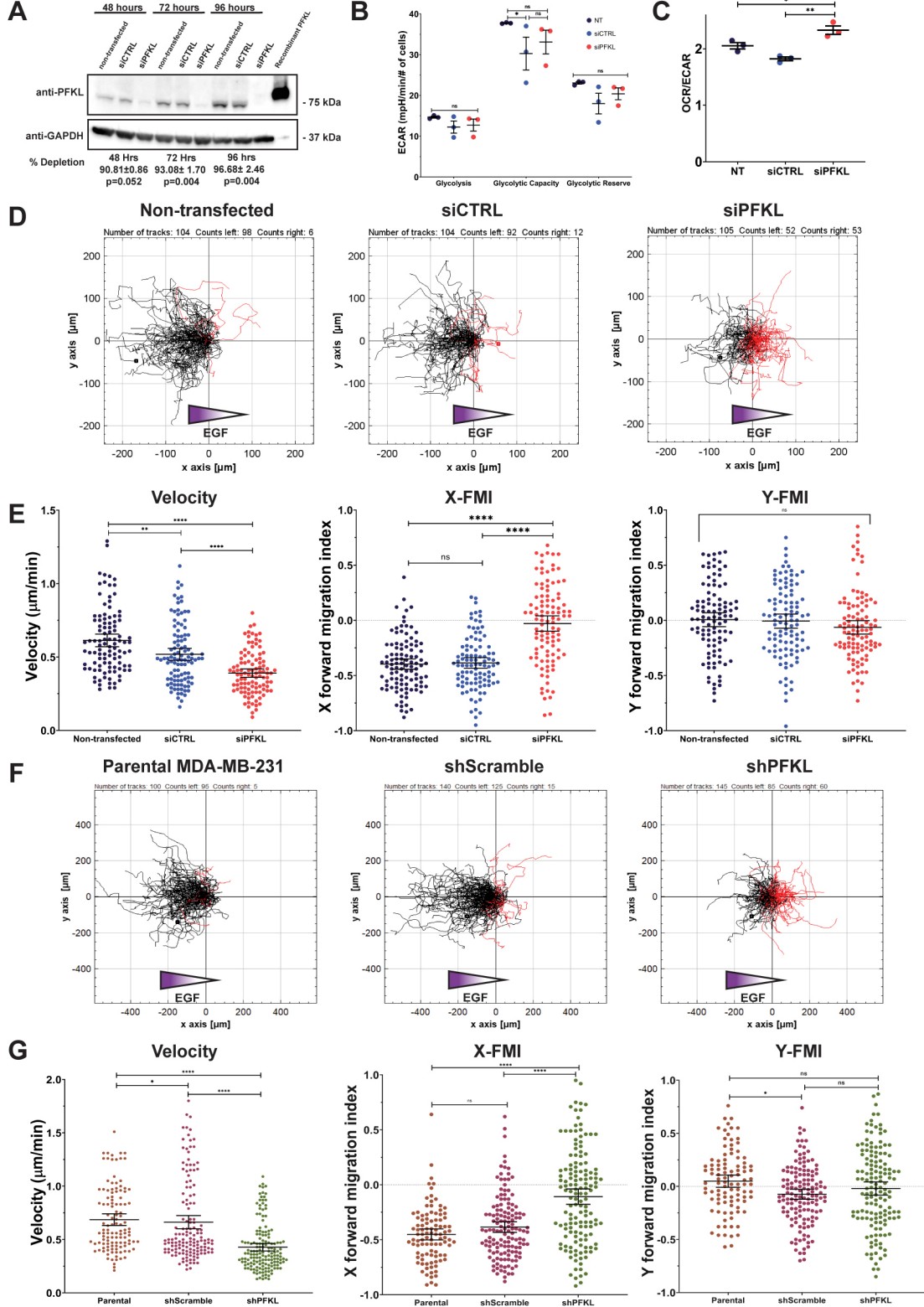

**Fig. 1.** See next page for legend.

to only 58.6% of shPFKL cells (Fig. 1F), indicating that PFKL-depleted cells migrated in a random fashion. PFKL shRNA-expressing cells had significant decreases in both migration velocity and directional sensing (Fig. 1G; Table S2), mirroring our findings with siRNA depletion. Together, these data show that PFKL is required to sustain directional migration in breast cancer cells.

## PFKL localizes to lamellipodia

Lamellipodia formation is essential for efficient chemotaxis of MDA-MB-231 cells on a two-dimensional surface. We next asked whether PFKL is localized to leading-edge lamellipodia of EGF-stimulated cells. In our hands, commercial PFKL antibodies in fixed-cell imaging experiments have non-specific signal, which

**Fig. 1. PFKL depletion decreases chemotaxis.** (A) Western blot of cell lysates of non-transfected (NT), siCTRL-transfected or siPFKL-transfected cells probed with antibodies raised against PFKL or GAPDH as a loading control. Recombinant PFKL protein (10 ng) was loaded as a positive control. Quantification of the percentage of PFKL depletion in siPFKL cells compared to siCTRL cells is shown below (mean±s.e.m. and *P*-value; two-tailed paired *t*-test), *n*=3 biological replicates. Cells transfected with PFKL siRNA show a significant decrease in PFKL protein 48–96 h after transfection. (B) The real-time assessment of ECAR for NT cells, siCTRL cells and siPFKL cells was performed using a Seahorse XFe96 Analyzer. Rate of glycolysis, glycolytic capacity and glycolytic reserve were determined. (C) The ratio of OCR to ECAR was determined for cells as in B. Data in B,C are *n*=3 and error bars represent s.e.m. *P<0.05; **P<0.01; ns, not significant (calculated by ordinary one-way ANOVA followed by Tukey's multiple comparisons test). (D) Spider diagrams visualizing single-cell tracks during chemotaxis assays of the indicated cell lines. Black tracks indicate migration up the EGF gradient, red tracks indicate migration down the gradient. (E) Quantification of cell tracking data as in D. The velocity of migration, and the migration parallel (X-FMI) and perpendicular (Y- FMI) to the EGF gradient is shown. In total, 104 (non-transfected and control siRNA-transfected) or 105 (PFKL siRNA-transfected) cells from three biological replicates were analyzed. Bars represent mean ±s.e.m. ****P<0.0001; **P<0.01; ns, not significant (calculated by ordinary one-way ANOVA followed by Tukey's multiple comparison test). (F) Spider diagrams visualizing single-cell tracks during chemotaxis assays of the indicated cell lines. Black tracks indicate migration up the EGF gradient, red tracks indicate migration down the gradient. (G) Quantification of cell tracking data as in F. The velocity of migration, and the migration parallel (X-FMI) and perpendicular (Y-FMI) to the EGF gradient is shown. In total, 100 parental, 140 shScramble (pLKO.1 Scramble) and 145 shPFKL (pLKO.1 767) cells from three biological replicates were analyzed for each condition. Bars represent mean±s.e.m. ****P<0.0001; *P<0.05; ns, not significant (calculated by ordinary one-way ANOVA followed by Tukey's multiple comparison test).

might be due to cross-reactivity with the other PFK1 isoforms. To overcome this challenge, we generated MDA-MB-231 cells stably expressing exogenous wild-type human PFKL with an amino-terminal FLAG tag using a lentiviral expression system. The expression of FLAG–PFKL was confirmed by western blot analysis (Fig. S1I). Localization of FLAG–PFKL to lamellipodia was assessed by immunofluorescence staining of serum-starved cells stimulated with EGF, which promotes lamellipodia formation, cell migration and metastasis of MDA-MB-231 cells (Biswenger et al., 2018; Price et al., 1999; Yang et al., 2011). FLAG–PFKL had a diffuse appearance in the cytoplasm, with distinct areas of enrichment in the lamellipodia, consistent with the ability of PFKL to form biomolecular condensates in the cytoplasm (Jin et al., 2017; Lynch et al., 2024; Webb et al., 2017) (Fig. 2). Next, we asked whether PFKL colocalized with other rate-limiting glycolytic enzymes, which are essential to produce ATP. We observed colocalization of PFKL with the rate-limiting enzymes hexokinase-2 (HK2) and pyruvate kinase M2 (PKM2) in lamellipodia (Fig. 2), where FLAG–PFKL colocalized with HK2 in 68.3±2.2% of cells (mean±s.e.m.; 73 cells in total, three biological replicates) and PKM2 in 73.9±3.1% of cells (mean±s.e.m.; 92 cells in total, three biological replicates). Together, these results support the hypothesis that PFKL forms a glycolytic metabolon in lamellipodia to locally generate ATP to support cell migration.

## PFKL catalytic activity is required for directional migration
We next sought to determine the functional requirements of PFKL in cell migration. First, we asked whether catalytic activity is required for chemotaxis. If our hypothesis is correct, we predict that loss of PFKL catalytic activity would decrease chemotaxis. To test this prediction, we expressed a previously characterized mutation, PFKL-H199Y, that inhibits enzyme activity but does not prevent

tetramer formation (Webb et al., 2015). Cells expressing FLAG–PFKL-H199Y were generated using a lentiviral expression system, and the expression of wild-type or mutant FLAG-tagged PFKL was confirmed by western blot analysis (Fig. 3A). EGFP was used as a control for transduction. FLAG–PFKL-H199Y was expressed in cells at approximately six times the levels of the endogenous protein in EGFP-transformed cells. Wild-type FLAG–PFKL showed a higher level of expression, with a PFKL level approximately 15 times that of control cells. To characterize the impact of exogenous PFKL expression on cellular metabolism, we estimated glycolytic flux by measuring ECAR. Expression of wild-type FLAG–PFKL increased the rate of glycolysis and decreased the OCR-to-ECAR ratio, consistent with a decreased reliance on mitochondrial activity (Fig. 3B; Fig. S2A,B). In cells expressing FLAG–PFKL-H199Y, a decrease in glycolytic flux and glycolytic capacity was observed. As the MDA-MB-231 cells express endogenous PFK1, these data are consistent with the catalytically inactive PFKL-H199Y acting as a dominant negative, decreasing glycolytic flux. Despite having drastically different glycolytic programs, no significant difference in proliferation was observed (Fig. S2C).

Next, we asked whether catalytic activity was required for lamellipodial localization of PFKL. Overall, the localization of FLAG–PFKL-H199Y was similar to that of the wild-type protein; FLAG–PFKL-H199Y was diffuse in the cytoplasm and present in lamellipodia (Fig. 3C). To determine whether PFKL was enriched in lamellipodia, we quantified the lamellipodial-to-cytoplasmic ratio of PFKL in line scans. FLAG–PFKL had an average ratio of 1.52±0.05 (mean±s.e.m.), indicating that the wild-type enzyme was present at greater levels in the lamellipodia compared to the cytoplasm. FLAG–PFKL-H199Y showed a significantly reduced enrichment in lamellipodia compared with the wild-type enzyme, with a lower ratio of 1.35±0.06 (*P*=0.04, two-tailed paired *t*-test; Fig. S2D). One explanation for the reduced lamellipodial enrichment is the significantly lower expression levels of the inactive enzyme. However, as the inactive PFKL-H199Y forms shorter filaments (Webb et al., 2017) and filament formation is required for recruitment to lamellipodia (see below), enzyme activity might be required for efficient localization of the enzyme to the cell periphery.

We next determined the impact of exogenous PFKL expression on cell migration (Fig. 3D,E; Table S3). Lentiviral transduction had little impact on cell migration, as the migration parameters of EGFP control cells were nearly identical to those of non-transfected controls (Tables S1, S3). Exogenous expression of FLAG–PFKL had little effect on migration velocity, in contrast to RNAi-mediated depletion, suggesting that basal, regulated PFKL levels are sufficient for its role. PFKL is subject to tight allosteric regulation, closely coupling its activity to cellular needs. Consistent with other proteins involved in the regulation of cell migration such as Rac1 and RhoA, which are required for motility but do not further enhance migration velocity when overexpressed, we propose that PFKL supports migration only above a threshold level of expression, beyond which additional expression has little impact due to allosteric regulation (Pankov et al., 2005; Sailland et al., 2014).

In contrast, FLAG–PFKL-H199Y expression significantly altered chemotaxis parameters, resembling the phenotype caused by RNAi-mediated PFKL depletion (Fig. 1). Specifically, only 53.2% of cells expressing FLAG–PFKL-H199Y showed tracking up the EGF gradient (Fig. 3D), indicating that the cells are migrating randomly rather than directionally towards the chemoattractant source. This corresponds to significant decreases in the velocity of migration and magnitude of X-FMI compared to those of EGFP control or wild-type PFKL-expressing cells (Fig. 3E). Together,

Journal of Cell Science

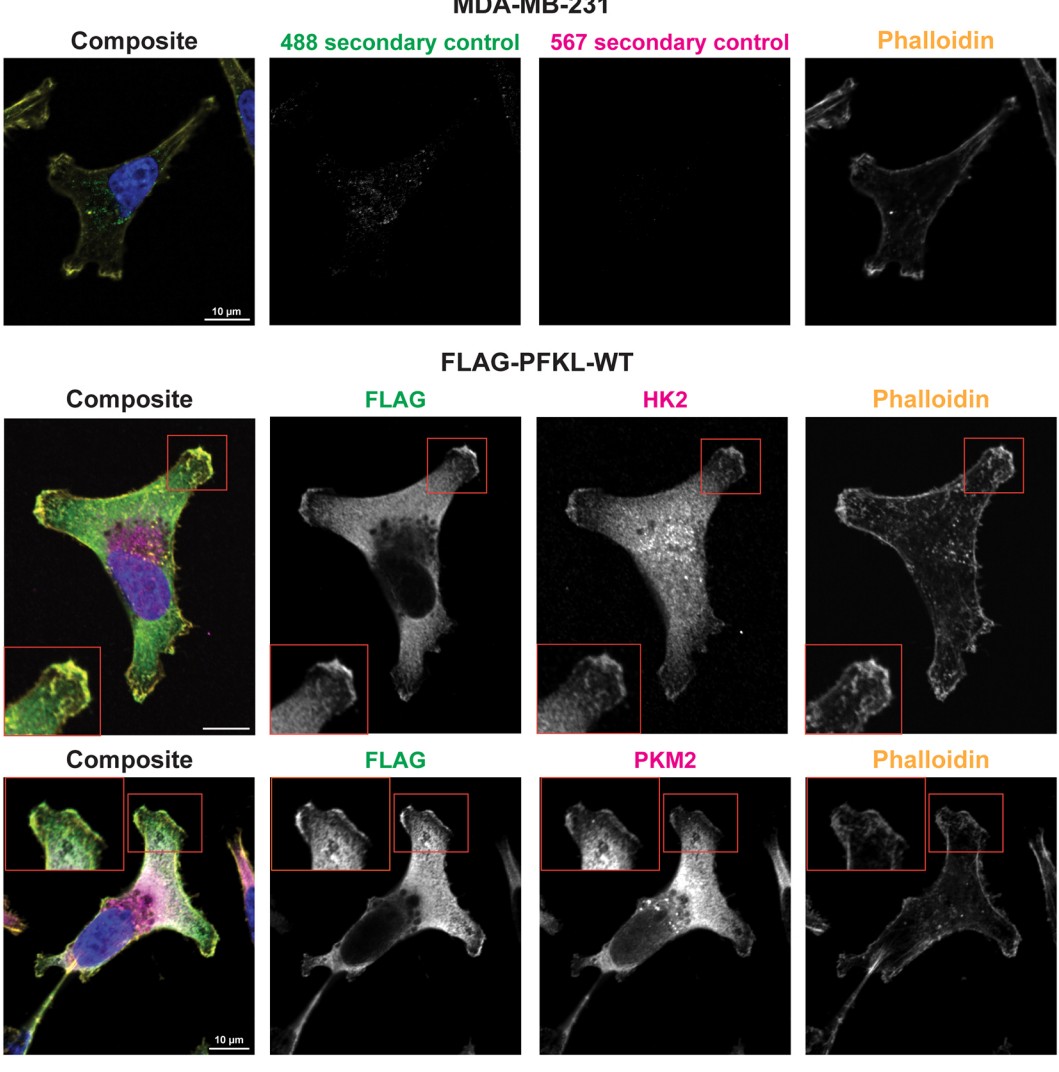

**Fig. 2. PFKL localizes to lamellipodia in MDA-MB-231 cells.** Fixed-cell imaging of cells stained with Hoechst 33342 (blue) and phalloidin–Alexa Fluor-647 (yellow). MDA-MB-231 cells were additionally labeled with Alexa Fluor-488- and Alexa Fluor-567-conjugated secondary antibodies as controls (top). In cells transduced with wild-type FLAG–PFKL (FLAG-PFKL-WT), the localization of PFKL was labeled with anti-FLAG M2 (green), along with labeling of either hexokinase 2 (HK2; magenta, middle) or pyruvate kinase M2 (PKM2; magenta, bottom). Boxes indicate regions shown as enlarged insets. Scale bars: 10 µm. Images shown are representative of three experiments.

these data show that PFKL kinase activity is required for directional migration.

### Inhibition of PFKFB3 decreases chemotaxis

We next asked whether chemotaxis of MDA-MB-231 cells was dependent on the production of fructose-2,6-bisphosphate (F-2,6-bP), a potent allosteric activator, by phosphofructokinase-2/fructose-2,6-bisphosphatase isoform 3 (PFKFB3). PFKFB3 is a bifunctional enzyme that has greater kinase activity compared to phosphatase activity and is upregulated in cancers. PFKFB3 is necessary for cancer cell migration and can be activated downstream of the EGF receptor (EGFR) by extracellular signal-regulated kinase (ERK), AMP-activated kinase (AMPK) or protein kinase A (PKA) signaling (Fig. 4A) (Shi et al., 2017). Treatment of cells with PFKFB3-specific inhibitor PFK15 decreases cell migration and proliferation in MDA-MB-231 cells (Kashyap et al., 2023). However, the impact on chemotaxis in MDA-MB-231 cells has not been shown. We first confirmed that PFKFB3 is expressed in MDA-MB-231 cells by western blot analysis, using recombinant

GST–PFKFB3 as a positive control (Fig. S2E). Next, we confirmed that PFK15 is able to inhibit the enzyme at the concentrations used to treat our cells. We modified our *in vitro* PFK1 auxiliary enzyme activity assay to link the production of ADP to NADH reduction (Voronkova et al., 2023). Recombinant PFKFB3 in the presence of DMSO was active, producing 1.5 µmoles F-2,6-bP per minute per milligram of enzyme (Fig. S2F). Reactions containing 2 µM PFK15 had ~50% of the enzyme activity of control wells, confirming that PFK15 is efficacious at this concentration. We next determined the impact PFK15 treatment has on chemotaxis. Treatment of cells with DMSO had little impact on cell migration, as no significant differences in measured parameters were observed compared to non-transfected cells (Fig. 4B,C; Table S4). Treatment with 2 µM PFK15, however, significantly altered directional migration. We found that 92.3% of tracked cells treated with DMSO migrated up the EGF gradient, compared to 79.1% of PFK15-treated cells, indicating that treatment with PFK15 increased random migration. Migration of PFK15-treated cells up the EGF gradient, as measured by the X-FMI, was significantly decreased compared to controls,

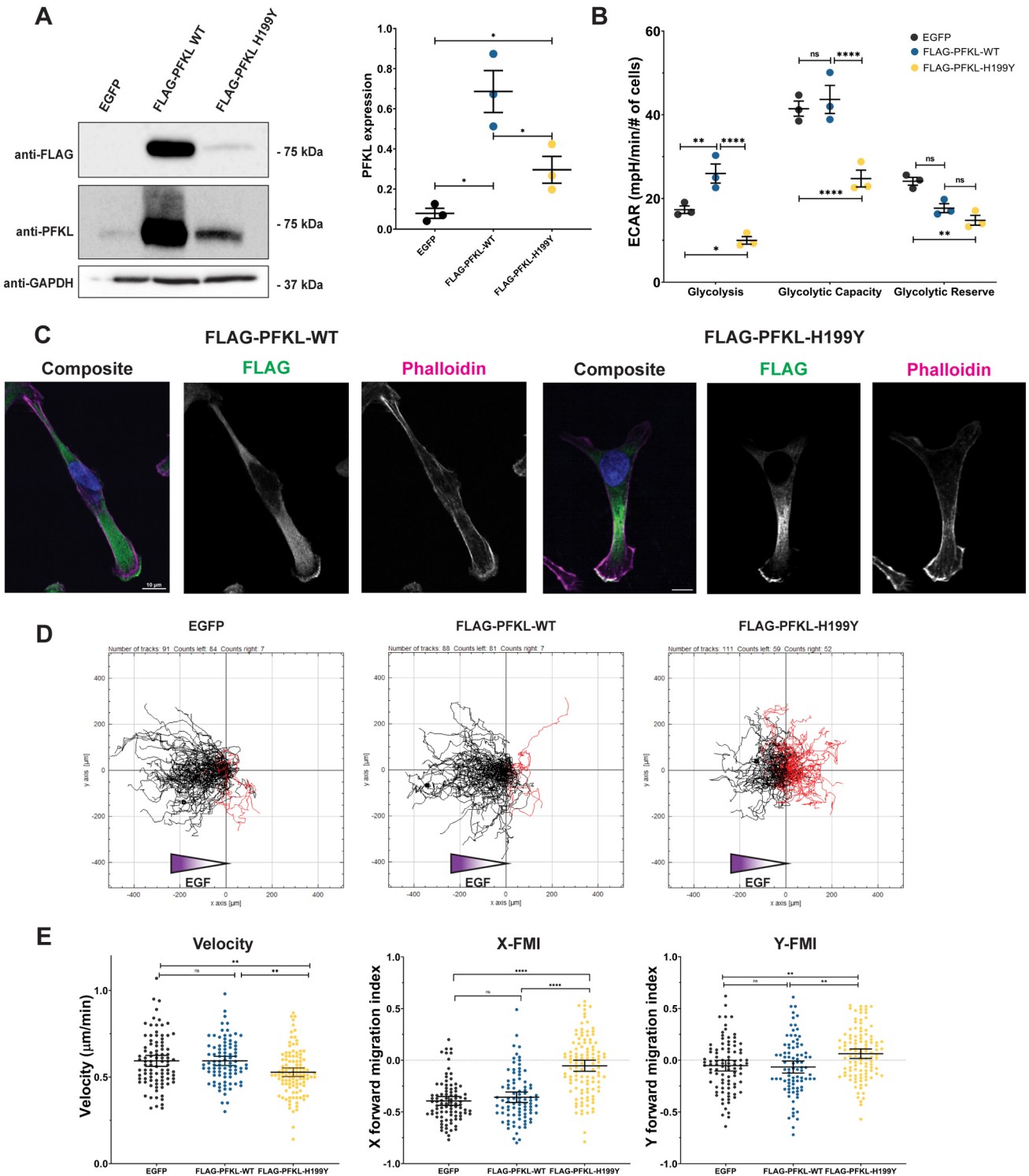

**Fig. 3. Catalytically inactive PFKL decreases directional migration.** (A) Western blot of cell lysates of MDA-MB-231 cells transduced with EGFP, wild-type (WT) FLAG–PFKL or FLAG–PFKL-H199Y, probed with antibodies raised against FLAG, PFKL or GAPDH (left). Quantification of PFKL expression relative to GAPDH loading control (right). $n$=3. Error bars represent s.e.m. *$P$<0.05 (significance determined using a two-tailed paired $t$-test). (B) Assessment of ECAR for the indicated cell lines was performed using a Seahorse XFe96 Analyzer. Rate of glycolysis, glycolytic capacity and glycolytic reserve were measured. Data are means of three independent experiments, and error bars represent s.e.m. *$P$<0.05; **$P$<0.01; ****$P$<0.0001; ns, not significant (calculated by ordinary one-way ANOVA followed by Tukey's multiple comparisons test). (C) Fixed-cell imaging of MDA-MB-231 cells transduced with WT FLAG–PFKL or FLAG–PFKL-H199Y and subsequently stained with Hoechst 33342 (blue), phalloidin–Alexa Fluor-647 (magenta) and anti-FLAG M2 (green). Scale bars: 10 µm. Images shown are representative of three experiments. (D) Spider diagrams visualizing single-cell tracks during chemotaxis assays of the indicated cell lines. Black tracks indicate migration up the EGF gradient, red tracks indicate migration down the gradient. (E) Quantification of cell tracking data as in D. The velocity of migration, and the migration parallel (X-FMI) and perpendicular (Y-FMI) to the EGF gradient is shown. In total, 91 (EGFP), 88 (WT FLAG–PFKL) or 111 (FLAG–PFKL-H199Y) cells from three biological replicates were analyzed. Bars represent mean±s.e.m. ****$P$<0.0001; **$P$<0.01; ns, not significant (calculated by ordinary one-way ANOVA followed by Tukey's multiple comparison test).

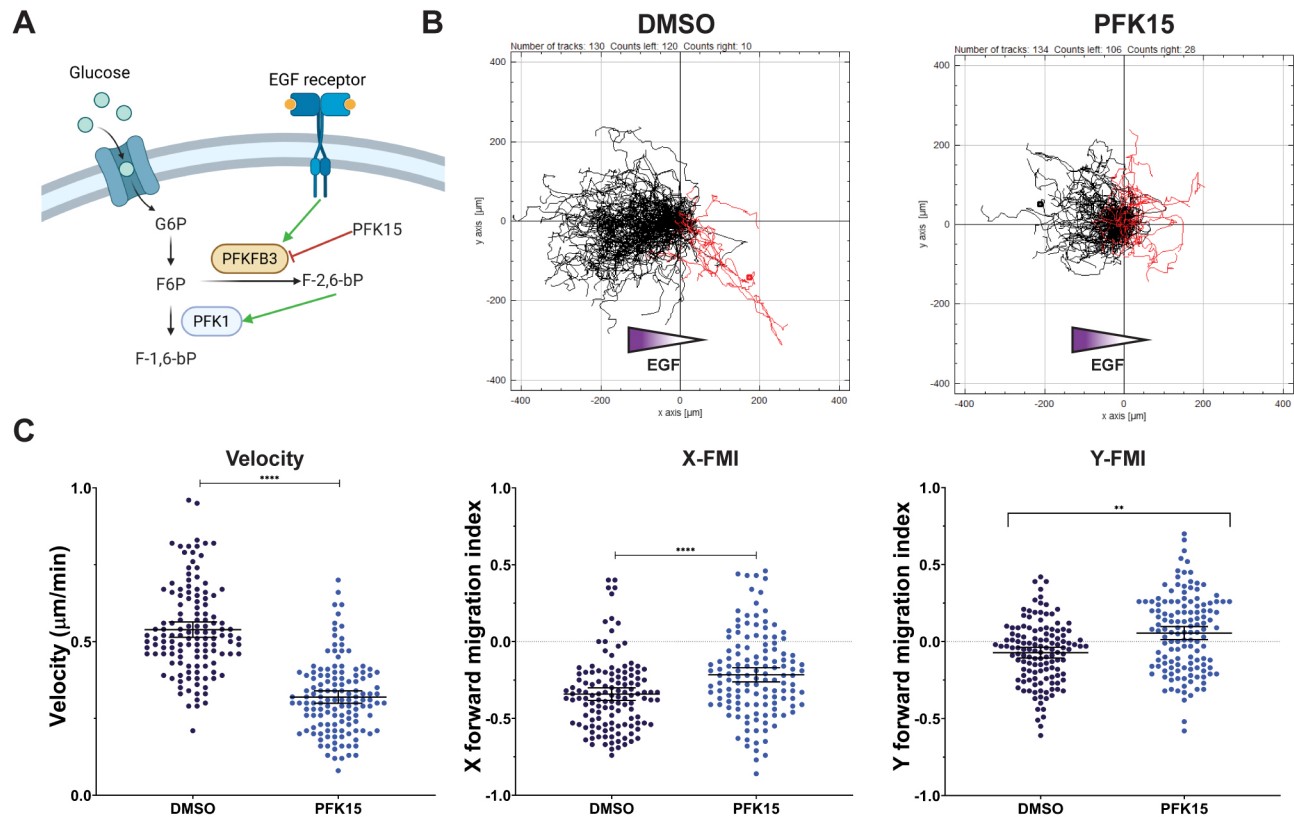

**Fig. 4. PFKFB3 activity is required for directional migration.** (A) Schematic depicting PFKL activation via PFKFB3. G6P, glucose-6-phosphate. (B,C) Spider diagrams (B) and quantification (C) of single-cell chemotaxis assays for control DMSO-treated MDA-MB-231 cells and PFK15-treated (2 μM) MDA-MB-231 cells. In B, black tracks indicate migration up the EGF gradient, red tracks indicate migration down the gradient. The velocity of migration, and the migration parallel (X-FMI) and perpendicular (Y-FMI) to the EGF gradient is shown. In total, 130 (DMSO) and 134 (PFK15) cells from three biological replicates were analyzed. Bars represent mean±s.e.m. ****$P$<0.0001, **$P$<0.01 (calculated by ordinary one-way ANOVA followed by Tukey's multiple comparison test).

but this effect was not as severe as that of siRNA depletion or expression of the kinase-inactive mutant. Further, we observed that cells treated with PFK15 had a significant decrease in migration velocity compared to that of the DMSO-treated control cells (DMSO, 0.53±0.14; PFK15, 0.31±0.12; mean±s.e.m.). These data are consistent with PFK1 activation by F-2,6-bP generated by PFKFB3 to support cell migration.

## PFKL filamentation is required for lamellipodial localization and directional sensing

Finally, we asked whether filament formation by PFKL is necessary for directional migration. A unique property of PFKL is the ability to form filaments of stacked tetramers through an electrostatic interaction between asparagine 702 of one tetramer with asparagine 702 of an adjacent tetramer (Lynch et al., 2024; Webb et al., 2017). We have previously shown that mutation of asparagine 702 to threonine renders PFKL unable to form filaments *in vitro* and assemble into condensates in human HepG2 hepatocellular carcinoma cells (Lynch et al., 2024). To ask whether filament formation alters the cellular distribution of PFKL in our present model, we generated MDA-MB-231 cell lines stably expressing filament-incompetent FLAG–PFKL-N702T using lentiviral transduction and confirmed expression using western blot analysis (Fig. 5A). To determine whether disruption of the filament interface altered glucose metabolism, we measured ECAR of cells expressing EGFP, wild-type FLAG–PFKL or filament-incompetent FLAG–PFKL-N702T. Cells expressing FLAG–PFKL-N702T behaved like the wild-type PFKL-expressing cells; a similar

increase in basal glycolytic rate and decrease in glycolytic reserve was observed in both cell lines compared to control EGFP-expressing cells (Fig. 5B; Fig. S3A,B). However, no difference was observed between cell expressing wild-type PFKL and cells expressing filament-incompetent PFKL. Additionally, no differences in proliferation were observed between EGFP control cells, FLAG–PFKL-expressing cells and FLAG–PFKL-N702T-expressing cells (Fig. S3C).

Immunofluorescence staining revealed a similar diffuse appearance in the cytoplasm for both wild-type and filament-incompetent FLAG–PFKL (Fig. 5C). However, FLAG–PFKL-N702T showed a significant decrease in levels in the lamellipodia compared to the wild-type protein. In cells stimulated with EGF to promote lamellipodia formation, we observed that 79.5%±3.5% (mean±s.e.m.) of FLAG–PFKL cells exhibited PFKL localization to lamellipodia, whereas lamellipodial localization was observed in only 30.4%±3.0% of cells expressing FLAG–FPKL-N702T, significantly less than wild type ($P$=4.9×10$^{-5}$, two-tailed paired *t*-test; Fig. S3D). To quantify the abundance of FLAG–PFKL in the lamellipodia, line scans were performed and the ratio of lamellipodial signal to cytoplasmic signal was calculated. FLAG–PFKL had an average ratio of 1.53±0.07 (mean±s.e.m.), indicating that lamellipodia had a higher FLAG–PFKL signal compared to the cytoplasm. Conversely, FLAG–PFKL-N702T cells had an average ratio of 0.94±0.05, showing decreased levels present in the lamellipodia compared to the cytoplasm ($P$=6.02×10$^{-10}$; Fig. 5D). These data suggest that filament-incompetent PFKL-N702T is not recruited to lamellipodia like wild-type PFKL. To verify these results, we performed a differential

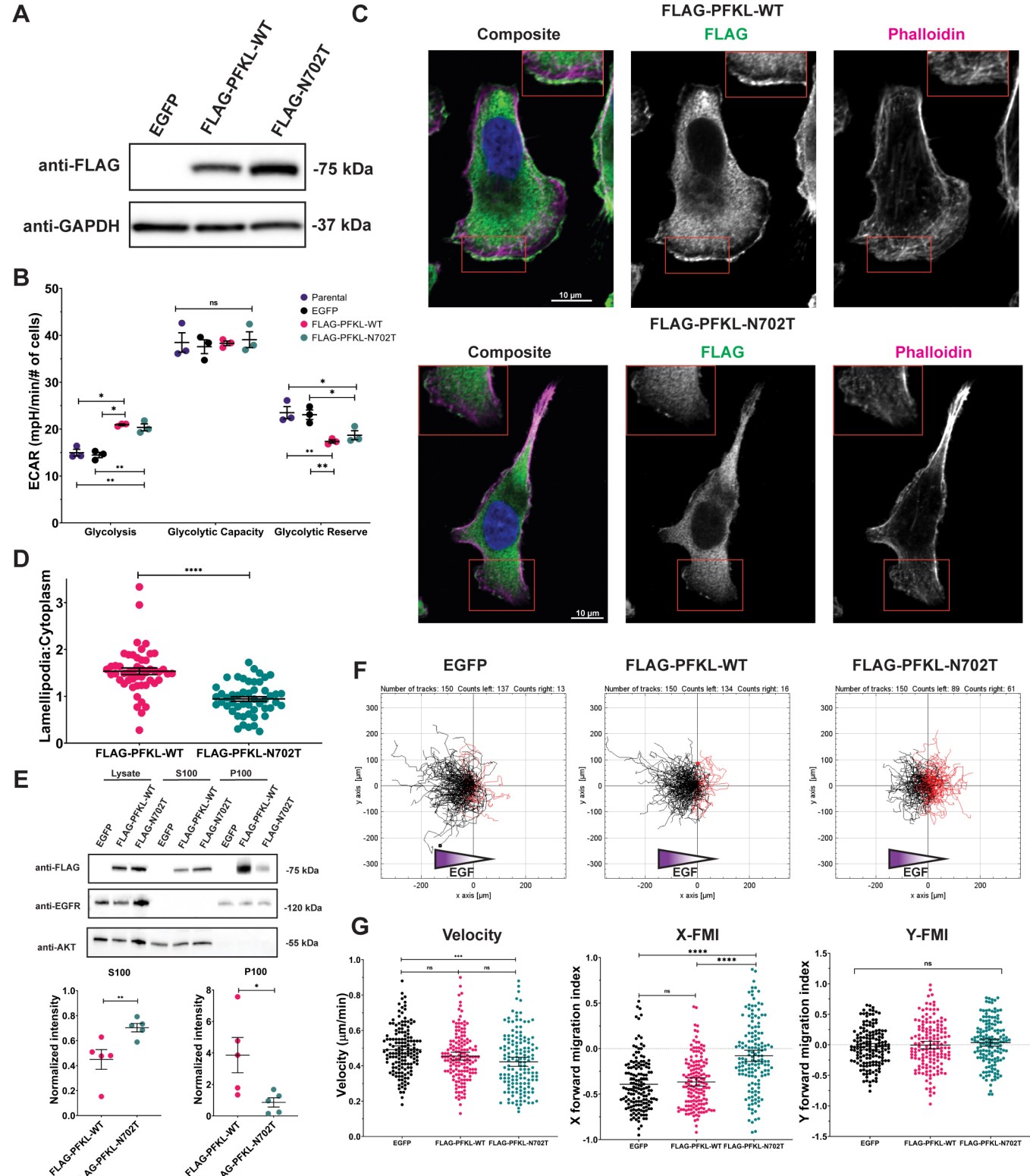

**Fig. 5.** See next page for legend.

cellular fractionation on post-nuclear supernatants to determine whether the enzymes reside in a 100,000 $g$ soluble fraction (S100), containing the supernatant, or in the insoluble fraction (P100), a crude particulate fraction containing the plasma membrane, organelles and the cytoskeleton (Hill and Hemmings, 2002). We confirmed that our fractionation protocol was able to separate soluble and insoluble

fractions by determining that the transmembrane EGFR was in the P100 fraction and the cytoplasmic AKT protein kinases were in the S100 fraction (Fig. 5E). Wild-type FLAG–PFKL was found in both the soluble and insoluble fractions, with higher abundance in the P100 fraction. The filament-incompetent FLAG–PFKL-N702T showed a significant decrease in the insoluble levels, with a majority found in

**Fig. 5. PFKL filament-mediated localization is required for directional sensing.** (A) Western blots of cell lysates of MDA-MB-231 cells transduced with EGFP, wild-type (WT) FLAG–PFKL or FLAG–PFKL-N702T, probed with antibodies raised against FLAG or GAPDH loading control. Blots shown are representative of three experiments. (B) Assessment of ECAR was performed for the indicated cell lines using a Seahorse XFe96 Analyzer. Rate of glycolysis, glycolytic capacity and glycolytic reserve were measured. Data are means of three independent experiments, and error bars represent s.e.m. *$P<0.05$; **$P<0.01$; ns, not significant (calculated by ordinary one-way ANOVA followed by Tukey's multiple comparisons test). (C) Fixed-cell imaging of MDA-MB-231 cells transduced with WT FLAG–PFKL or FLAG–PFKL-N702T and subsequently stained with Hoechst 33342 (blue), phalloidin–Alexa Fluor-647 (magenta) and anti-FLAG M2 (green). Boxes indicate lamellipodial regions shown as enlarged insets. Scale bars: 10 µm. (D) Quantification of 50 representative FLAG–PFKL and FLAG–PFKL-N702T cells as in C from four independent experiments using line scans spanning lamellipodia and cytoplasm. Bars represent mean±s.e.m. ****$P<0.0001$ (significance calculated using a two-tailed unpaired $t$-test). (E) Western blot of lysate and of S100 and P100 fractions, separated by subcellular fractionation, from the indicated cell lines. The blot was probed with antibodies raised against FLAG, EGFR and AKT kinases (top). Quantification comparing FLAG–PFKL and FLAG–PFKL-N702T presence in S100 and P100 fractions, normalized to fraction controls. $n=5$. Error bars represent s.e.m. *$P<0.05$; **$P<0.01$ (significance was determined using a two-tailed paired $t$-test). (F) Spider diagrams visualizing single-cell tracks during chemotaxis assays of the indicated cell lines. Black tracks indicate migration up the EGF gradient, red tracks indicate migration down the gradient. (G) Quantification of cell tracking data as in F. The velocity of migration, and the migration parallel (X-FMI) and perpendicular (Y-FMI) to the EGF gradient is shown. In total, 150 cells from three biological replicates were analyzed for each condition. Bars represent mean±s.e.m. ****$P<0.0001$; ***$P<0.001$; ns, not significant (calculated by ordinary one-way ANOVA followed by Tukey's multiple comparison test).

the S100 fraction. These data support our findings from Fig. 5C,D, indicating that PFKL filament formation alters PFKL localization within the cytoplasm.

We next asked whether filament formation was required for chemotaxis. EGFP control cells and wild-type FLAG–PFKL-expressing cells had a strong directional preference towards the EGF gradient, with a large majority (91.3% and 89.5%, respectively) of cells migrating up the gradient. In contrast, cells expressing filament incompetent FLAG–PFKL-N702T had a significantly reduced directional migration, with only 59.3% of cells moving up the gradient, indicating that overall migration was more random than directional. There was no difference in X-FMI between cells expressing EGFP or FLAG–PFKL, whereas cells expressing FLAG–PFKL-N702T had a significant decrease in magnitude of X-FMI (Fig. 5F,G; Table S5). Interestingly, there was no significant difference in migration velocity between cells expressing wild-type or filament-incompetent FLAG–PFKL (Fig. 5G). Taken together, these data suggest that PFKL filaments are important to localize the enzyme to the lamellipodia, where it supports a role in directional cell migration.

## DISCUSSION
Here, we investigated the role of the liver isoform of PFK1 in chemotaxis of MDA-MB-231 human breast cancer cells. We show that PFKL is required for chemotaxis, as RNAi-mediated depletion significantly decreased the velocity of migrating cells and the ability of cells to migrate up an EGF gradient. Further, whereas overexpression of wild-type PFKL did not impact directional migration, overexpression of a catalytically inactive PFKL mutant or inhibition of F-2,6-bP production decreased both migration velocity and directional sensing, indicating that catalytic activity is required for chemotaxis. Finally, we found that disrupting PFKL

filament formation altered the subcellular localization of PFKL, reducing recruitment to the plasma membrane and directional sensing. Together, these results demonstrate the requirements of PFKL in chemotaxis.

Our work adds to a growing body of research demonstrating the importance of the localization of metabolic enzymes, such as glycolytic enzymes, in supporting cellular functions (Fukuhara et al., 2025). In the cell body, a PFKL-containing metabolon localizes to mitochondria, where it forms a functional metabolic network to efficiently generate ATP (Kyoung et al., 2024). Additionally, PFKL can form biomolecular condensates in the cell body in response to hypoxic stress (Jin et al., 2017), serum starvation (Lynch et al., 2024) and citrate stimulation (Sivadas et al., 2023; Webb et al., 2017), as well as during cell cycle progression (Jeon et al., 2023). Localized glycolysis also supports energy-intensive processes at the plasma membrane, including cytoskeletal remodeling, formation of endothelial cell protrusions and synaptic activity under energetic stress (De Bock et al., 2013; DeWane et al., 2021; Jang et al., 2016; Park et al., 2020; Santos et al., 2023). We show here that filament formation by PFKL is required to localize the enzyme to lamellipodia (Fig. 5C–E). Filament formation by intermediary metabolic enzymes has been found in enzymes from diverse pathways, including nucleotide metabolism (e.g. CTP synthase), lipid metabolism (e.g. acetyl-CoA carboxylase) and amino acid metabolism (e.g. glutamine synthase) (Lynch et al., 2020). Filament formation regulates enzyme activity through multiple mechanisms, including altering allosteric regulation, trapping toxic metabolic intermediates, acting as a signaling scaffold and altering the shape of cells (Beaty and Lane, 1983; Frey et al., 1975; Hansen et al., 2021; Lynch and Kollman, 2020; Meredith and Lane, 1978; O'Connell et al., 2012; Park and Horton, 2019; Shen et al., 2016). However, the filament-dependent function of PFKL in lamellipodia remains unclear. As PFKL colocalizes with other rate-limiting glycolytic enzymes (Fig. 2), PFKL filaments could enable the formation of a glycolytic metabolon to rapidly produce ATP in the lamellipodia. Alternatively, PFKL filaments could possess non-glycolytic functions, such as acting as a scaffold for plasma membrane-associated processes. For example, PFKP monomers have been shown to have a scaffolding function in EGFR signaling (Lee et al., 2017). Lastly, PFKL filaments could act as a metabolic sensor and participate in cell signaling cascades by promoting the formation of glycolytic intermediates such as dihydroxyacetone phosphate (DHAP) or F-1,6-bP, which have been shown to act as second messengers in cell signaling (Icard et al., 2021; Li et al., 2024; Lim et al., 2016; Orozco et al., 2020; Peeters et al., 2017). F-1,6-bP has been shown to act as a cofactor for the *Saccharomyces cerevisiae* Ras guanine-nucleotide-exchange factor Cdc25 and its human ortholog Son of Sevenless 1 (SOS1) (Peeters et al., 2017). Ras activation is essential for directional cell migration, through activation of multiple signaling pathways including the phosphoinositide 3-kinase, ERK and mechanistic target of rapamycin complex 2 (mTORC2) pathways (Collins et al., 2023; Samson et al., 2022; Sepe et al., 2013; Zhan et al., 2020). Determining the molecular mechanism enabling directional sensing is an area of future research.

In contrast to directional sensing, loss of lamellipodial localization of PFKL did not alter the migration velocity of cells. Cells expressing FLAG–PFKL-N702T, which is primarily localized to the cytoplasm, maintain overall glycolytic flux and migration velocity but not directional sensing. These data suggest that a filament-independent mechanism of forming ATP in the cytosol is required to maintain migration velocity. In contrast, expression of PFKL-H199Y, which forms shorter filaments but is still present in

lamellipodia, significantly decreases overall glycolytic flux, migration velocity and directional sensing. These results highlight the necessity and different roles of PFKL in directional migration based on localization of the enzyme within the cell. Future studies using forced membrane localization of filament-incompetent PFKL are needed to clarify the role of PFKL in lamellipodia.

Several questions remain unresolved but are important to understand the role of localized glycolysis in chemotaxis. First, do other PFK1 isoforms contribute to the directional migration of breast cancer cells? To directly assess the role of PFK1 isoforms, isoform-specific PFK1 inhibitors are needed. Although ongoing efforts aim to generate these essential tools, no commercial inhibitors are currently available. While PFK15 significantly impairs both directional sensing and migration velocity, PFK15 inhibits PFK1 activity in an indirect, non-isoform-specific manner. PFK15 inhibits PFKFB3-dependent generation of F-2,6-bP, which is a potent allosteric activator of all three human PFK1 isoforms. Second, how are PFKL filaments are recruited to lamellipodia? Glycolytic enzymes, including PFK1, have been shown to associate with the actin cytoskeleton (Real-Hohn et al., 2010; Roberts and Somero, 1989, 1987; Sivadas et al., 2023). The PFKM–actin interaction is the most well characterized, and it is hypothesized that actin binding stabilizes the enzyme and increases activity (Kuo et al., 1986; Liou and Anderson, 1980). The effect of actin binding on PFKL and whether this is influenced by PFKL filaments, however, remains unclear. Glycolytic enzymes also associate with transmembrane proteins. For example, PFKP directly interacts with EGFR (Lee et al., 2017) and PFKM interacts with the transmembrane anion exchanger Band 3.1 (encoded by *SLC4A1*; Campanella et al., 2005; Chu et al., 2012; Chu and Low, 2006). Our preliminary results suggest that PFKL interacts with the sodium/hydrogen exchanger NHE1 in a filament-dependent manner. Although the significance of this interaction is under current investigation, NHE1 is enriched at the leading edge of migrating cells and is necessary for directional cell migration (Chiang et al., 2008; Zhang et al., 2022b). Finally, PFK1 has been shown to directly interact with phospholipids (Catimel et al., 2009, 2008), suggesting a potential lipid-dependent mechanism for localizing glycolysis to specific membrane domains analogous to phosphatidylinositol (3,4,5)-trisphosphate (PIP3)-dependent recruitment of AKT kinases to lamellipodia (Haugh et al., 2000). Third, at what stage of cell migration is PFKL recruited to the lamellipodia? Based on our present results, it is unclear whether PFKL is recruited to lamellipodia prior to EGF-stimulated protrusions or if it is recruited to protrusions that have already formed. Future research into the mechanism of recruitment will be essential to further our understanding of the role that localized glycolysis plays in chemotaxis.

Collectively, our findings provide insight into the function of PFKL in chemotaxing breast cancer cells. These results provide mechanistic understanding of breast cancer cell chemotaxis, which might provide insight into therapeutics designed to disrupt cancer metastasis. Future studies will investigate how filaments localize PFKL to lamellipodia, and the mechanism by which they contribute to directional cell migration.

## MATERIALS AND METHODS
### Cell culture, generation of stable cell lines and cell proliferation
MDA-MB-231 cells were purchased from ATCC and were verified to be mycoplasma free by using LookOut Mycoplasma PCR Detection Kit (Sigma-Aldrich, St. Louis MO, USA; MP0035-1KT). Cells were maintained in high-glucose DMEM (Thermo Fisher Scientific, Rockfield

IL, USA; 11965-092) supplemented with 10% heat-inactivated fetal bovine serum (FBS; Gibco, Grand Island NY, USA; A52568-01) and 1% penicillin-streptomycin (Gibco, Grand Island NY, USA; 15140-01) in a 37°C tissue culture incubator at 5% $CO_2$. MDA-MB-231 cells expressing human FLAG–PFKL or FLAG–PFKL-N702T were generated as previously described (Lynch et al., 2024). A pLV vector containing FLAG–PFKL with the H199Y point mutation (Webb et al., 2017) was purchased from Vector Builder. Lentivirus particles were generated, and MDA-MB-231 cells were infected as previously described (Voronkova et al., 2023). Briefly, cells were transduced with lentiviral particles encoding FLAG–PFKL. Antibiotic-resistant cells were pooled and used in cell assays. The expression of FLAG–PFKL and the mutants was verified by western blot analysis. Cell proliferation was determined by counting the number of cells in a 12-well plate 24–96 h after plating at an initial density of $0.11 \times 10^6$ cells per well. At each time point, cells were washed with PBS pH 7.4 (Gibco, Grand Island NY, USA; 10010-023) lifted from the plate by incubation with 0.25% trypsin (Gibco, Grand Island NY, USA; 25200-056), and the number of cells estimated by counting on a hemocytometer. Cells suspensions were mixed at a 1:1 ratio with 0.4% Trypan Blue (Gibco, Grand Island NY, USA; 72-57-1) to quantify cell viability.

### Immunofluorescence staining and analysis
Immunofluorescence analysis was performed as previously described (Lynch et al., 2024). MDA-MB-231 cells were plated at a density of $0.2 \times 10^6$ cells per well of a 6-well plate containing 1.5 mm coverslips in complete growth medium. A day after seeding, cells were serum starved by incubating in growth medium containing 0.1% heat-inactivated FBS for 16 h. Cells were rinsed twice in PBS and the medium replaced with fresh starvation medium. Cells were stimulated with 50 ng/ml human recombinant epidermal growth factor (Thermo Fisher Scientific, Rockfield IL, USA; PHG0311L) for 5 min, washed three times in PBS and fixed in 4% paraformaldehyde for 30 min. Cell were permeabilized in 0.1% Triton X-100 for 10 min, washed in PBS and blocked in blocking buffer [3% bovine serum albumin (Gold Biotechnology, St. Louis MO, USA; 9048-46-88), 1% normal goat serum (Biosynth, Louisville KY, USA; 88NG22S), 1% cold-water fish gelatin (Aurion Biotech, Seattle WA, USA; 900.033) in PBS] for 30 min. Primary antibodies to FLAG-M2 (Sigma-Aldrich, St. Louis MO, USA; F1804; 1:1000), hexokinase 2 (GeneTex, Irvine CA, USA; GTX111525; 1:1000) or pyruvate kinase M2 (Cell Signaling Technologies, Danvers MA, USA; 4053; 1:500), diluted in blocking buffer, were added, and coverslips were incubated at 37°C for 2 h. Coverslips were washed three times in PBS and incubated with goat anti-mouse IgG or anti-rabbit IgG secondary antibodies labeled with Alexa Fluor-488 (Thermo Fisher Scientific, Rockfield IL, USA; A-11001; 1:500), goat anti-rabbit IgG secondary antibody labeled with Alexa Fluor-567 (Thermo Fisher Scientific, Rockfield IL, USA; A11031; 1:500), Hoechst 33342 (Thermo Fisher Scientific, Rockfield IL, USA; H3570) and phalloidin–Alexa Fluor-647 (Thermo Fisher Scientific, Rockfield IL, USA; A22287) for 30 min at 37°C. Coverslips were washed three times in 1 ml PBS, rinsed in distilled deionized water and mounted on glass slides using Prolong Gold antifade reagent (Invitrogen, Carlsbad CA, USA; P36934). Coverslips were imaged using a Nikon TI2-E Inverted Microscope with a CSU-W1 Confocal Scanner spinning disk with a 60× oil immersion objective. Image processing was performed in NIS Elements and FIJI ImageJ (Schindelin et al., 2012). To estimate the abundance of FLAG–PFKL in the lamellipodia versus cytoplasm, line scans of maximum-intensity projections were generated in FIJI ImageJ, and the ratio of lamellipodia to cytoplasm intensity was quantified in Microsoft Excel.

### RNA interference depletion of PFKL
PFKL siRNA depletion was performed using a pool of four siRNAs targeting human PFKL (L-006822-00-0005), or a pool of four non-targeting siRNAs (D-001810-10-05) as a control (Horizon Discovery Inc, USA). MDA-MB-231 cells were plated at a density of 60,000 cells per well in a 24-well plate. At 24 h after seeding, cells were transfected with 5 pmol PFKL siRNA or control siRNA using Lipofectamine RNAiMAX transfection reagent (Invitrogen, Carlsbad CA, USA; 13778-075), according to the manufacturer's recommendations. At 24–96 h after transfections, cells were washed in PBS and lysed with Modified RIPA

lysis buffer [50 mM tris(hydroxymethyl)aminomethane-HCl (Tris-HCl pH 7.5), 150 mM sodium chloride, 1% IGEPAL CA630 (MP, Solon OH, USA; 9036-19-5), 1 mM ethylenediaminetetraacetic acid (EDTA)]. PFKL abundance was determined using western blot analysis. Assays were performed in triplicate. Protein levels were estimated via densitometry analysis using FIJI ImageJ.

pLKO.1 vectors containing control scramble shRNA (target sequence, 5′-CCGCAGGTATGCACGCGT-3′; Addgene, Watertown MA, USA; Addgene plasmid 1864; Sarbassov et al., 2005) or shRNA against the PFKL 3′ UTR (TRCN0000199860; Target sequence, 5′-GCCAGCCCTTGCTCTACCT-GG-3′; Sigma-Aldrich, St. Louis MO, USA) were purchased. Lentivirus particles were generated and MDA-MB-231 cells infected as previously described (Voronkova et al., 2023). Depletion of PFKL was verified by western blot analysis and quantified as described above.

## Western blot analysis

Cells were rinsed two times in PBS and lysed in Modified RIPA lysis buffer [50 mM tris(hydroxymethyl)aminomethane-HCl (Tris-HCl pH 7.5), 150 mM sodium chloride, 1% IGEPAL CA630, 1 mM EDTA] on ice for 10 min. The lysates were collected by scraping, transferred to 1.5 ml Eppendorf tubes, and insoluble debris pelleted by centrifugation 21,000 $g$ for 15 min at 4°C. The supernatant was transferred to a new tube, and protein concentration was determined using the Pierce BCA Protein Assay Kit (Thermo Fisher Scientific, Rockfield IL, USA; A55865). SDS-PAGE was performed on a 10% Tris-Glycine gel (Invitrogen, Carlsbad, CA, USA; XP00100BOX) with 10 μg of lysate and transferred onto a 0.45 μm PVDF membrane. Membranes were washed in Tris-buffered saline pH 7.5 with 0.1% Tween-20 (TBST) and blocked for 1 h in 5% non-fat milk dissolved in TBST. Membranes were incubated overnight at 4°C in primary antibodies for FLAG (Sigma-Aldrich, St. Louis MO, USA; F1804; 1:1000), PFKL (Abcam, Waltham MA, USA; ab181064; 1:1000), PFKM/P (Invitrogen, Carlsbad CA, USA; PA5-29366, 1:1000), PFKFB3 (Abcam, Waltham MA, USA; ab181861; 1:1000), EGFR (Invitrogen, Carlsbad CA, USA; PA1-1110; 1:1000), pan-AKT (Thermo Fisher Scientific, Rockfield IL, USA; 44-609G; 1:1000) or GAPDH (Cell Signaling Technologies, Danvers MA, USA; 5147; 1:2000). The primary antibody was removed, and membranes were washed for 5 min with TBST three times. The membranes were then incubated with anti-mouse IgG (Jackson ImmunoResearch, West Grove PA, USA; 715-035-150; 1:20,000) or anti-rabbit IgG (Jackson ImmunoResearch, West Grove PA, USA; 711-035-152; 1:10,000) secondary antibody in 5% non-fat milk (RPI, Mt Prospect IL, USA; M17200-500.0) in TBST for 1 h at room temperature. The membranes were rinsed for 5 min with TBST three times and imaged with West Femto Chemiluminescent SuperSignal (Thermo Fisher Scientific, Rockfield IL, USA) on a G-box gel imager. All original, uncropped blots are located in Fig. S4: lanes are numbered and antibody used, as well as molecular weight of protein of interest, is noted next to each blot. Orange boxes indicate which lanes were chosen to use in figures. If no lanes are boxed, this indicates all were used in the figure.

## Metabolic analysis

The total PFK1 activity of cell lysates was determined as previously described (Webb et al., 2015). Briefly, cells were lysed in ice-cold lysis buffer [10 mM potassium phosphate, 0.1% Triton X-100 and 1% EDTA-Free Protease Inhibitor Cocktail (Abcam, Waltham MA, USA, ab270055)] on ice for 10 min, 48 h after seeding in a 6-well plate. The lysates were collected by scraping, transferred to a 1.5 ml Eppendorf tube, and centrifuged at 4°C for 10 min to remove cellular debris. The concentration of the lysates was determined using the Pierce BCA Protein Assay Kits (Thermo Fisher Scientific, Rockfield IL, USA; A55865). Activity assays were performed in triplicate in a 200 μl reaction mixture containing 10 μg lysate, 0.45 units/ml aldolase (Sigma-Aldrich, St. Louis MO, USA; 9024-52-6), 4.5 units/ml triosephosphate isomerase (Sigma-Aldrich, St. Louis MO, USA; 9023-78-3) and 1.5 units/ml glycerol phosphate dehydrogenase (Sigma-Aldrich, St. Louis MO, USA; 9075-65-4), 50 mM 4-(2-hydroxyethyl)piperazine-1-ethanesulfonic acid (HEPES) pH 7.5 (Gold Biotechnology, St. Louis MO, USA; 75277-39-3), 100 mM potassium chloride, 1 mM dithiothreitol (DTT), 0.15 mM NADH, 5 mM ammonium sulfate, 0.25 mM ATP and 4 mM F6P in 96-well plate. The reactions were equilibrated to 25°C for 5 min prior to the

addition of 10 mM MgCl$_2$ to initiate the reaction. Wells containing no F6P were used to correct for PFK1-independent hydrolysis of NADH. The absorbance at 340 nm was measured at 15 s intervals using a Varioskan LUX Multimode microplate reader (Thermo Fisher Scientific, Rockfield IL, USA). One unit of activity was defined as the amount of enzyme that catalyzed the formation of 1 μmol of F-1,6-bP/min/mg lysate.

Lactic acid excretion assays were performed as previously described (Webb et al., 2015). Briefly, 48 h after seeding in a 6-well plate, cells were washed twice with PBS and incubated for 1 h with serum-free DMEM (Gibco, Grand Island NY, USA; 11965-092) supplemented with 5 mM glucose. A sample of the medium was centrifuged at maximal speed at 4°C for 10 min to pellet cellular debris. Analysis of each cell line and control cell-free was performed in triplicate. 50 μl of the centrifuged medium was transferred to a well of a 96-well plate. 100 μl Reagent A (300 mM hydrazine, 200 mM glycine, pH 9.5, and 20 mM NAD$^+$) and 50 μl Reagent B [200 U/ml LDH (Sigma-Aldrich, St. Louis MO, USA; 9001-60-9) in water] was added to the sample and standard wells and incubated for 1 h at room temperature. The absorbance at 340 nm was measured using a Varioskan LUX Multimode microplate reader (Thermo Fisher Scientific, Rockfield IL, USA) and was converted to the concentration of lactate using the standard curve. Concentrations were estimated by comparing absorbance to a standard curve of lactate (Sigma-Aldrich, St. Louis MO, USA; 71718-10G) diluted in medium.

Real-time metabolic analysis was performed using the Seahorse XF Glycolysis Stress Test Kit on a Seahorse XFe96 Analyzer (Agilent Technologies, Santa Clara CA, USA), as previously described (Voronkova et al., 2023). Cells were plated onto Seahorse XF96 microplate and allowed to attach overnight. Prior to the assay, cells were washed and the media replaced with XF DMEM base medium supplemented with 2 mM glutamine and cell equilibrated for 1 h in a non-carbon dioxide incubator at 37°C. The oxygen consumption rate (OCR) and extracellular acidification rate (ECAR) were measured through sequential injection of 10 mM glucose, 1 μM oligomycin and 100 mM 2-deoxy-glucose (2-DG). The data were normalized to cell number measured by a CyQUANT Cell Proliferation Assay (Thermo Fisher Scientific, Rockfield IL, USA; C7026) according to the manufacturer's recommendations. Analysis was performed with the Agilent Seahorse Wave Desktop software. Significance for each glycolytic parameter was determined using one-way ANOVA in Graphpad Prism software (Graph Pad Software version 8, Graph Pad Software Inc., CA, USA).

## PFKFB3 kinase assay

The efficacy of PFK15 treatment was measured using a pyruvate kinase/LDH kinetic plate reader assay, with modifications to our previously described PFK1 activity assay (Voronkova et al., 2023). Briefly, PFKFB3 activity was measured in a buffer containing 50 mM EDTA (pH 7.5), 100 mM potassium chloride, 1 mM DTT, 0.25 mM NADH, 0.3 mM phosphoenolpyruvic acid, 0.1 mM ATP, 2.5 mM F6P. The assay was performed with 25 ng of recombinant PFKFB3 (SinoBiological, Chesterbrook PA, USA; 501618437) per 200 μl reaction in the presence of 2 μM PFK15 or DMSO. Absorbance at 340 nm was measured using a Varioskan LUX Multimode microplate reader (Thermo Fisher Scientific, Rockfield IL, USA) and assays were performed in triplicate.

## Single-cell chemotaxis and quantification

Single-cell migration of MDA-MB-231 cells was evaluated using μ-slide chemotaxis chambers from ibidi USA, Inc., according to their established protocol (Biswenger et al., 2018; Zengel et al., 2011). 6 μl of cells was plated at a density of ~3×10$^6$ cells/ml in the middle reservoir of the chamber. Cells were allowed to adhere for 6 h before filling both winged chambers with low-serum medium. EGF (Thermo Fisher Scientific, Rockfield IL, USA; PHG0311L) was added to the left chamber (in the negative-$x$ direction) with a final concentration of 50 ng/ml. The chemotaxis slide was incubated in the stage-top incubator for ~20 min before beginning the assay. Brightfield images of 4–5 fields of view of each chamber were captured using a 20× objective. Images were captured every 10 min for 16 h. Representative cells (~35–50) were imaged from each experimental group for $n$=3 biological replicate. Cells were tracked using the ImageJ manual tracking plugin. Analysis of cell tracks was performed using the Ibidi Chemotaxis and Migration Tool ImageJ plugin. Analysis of velocity, X-FMI and Y-FMI was performed using averages of each

single-cell value for each parameter. Statistical significance was determined by one-way ANOVA using GraphPad Prism.

## Cell fractionation

Cell fractionation was performed as previously described (Frantz et al., 2007; Buchan et al., 2002) with modifications. Briefly, MDA-MB-231 cells stably expressing EGFP, FLAG–PFKL or FLAG–PFKL-N702T were plated in 100 mm dishes. At 70% confluency, cells were washed two times with STE buffer (150 mM NaCl, 50 mM Tris-HCl pH 7.5, 1 mM EDTA). Cells were lysed on ice in 500 µl Hypotonic Buffer (10 mM Tris-HCl pH 7.5, 0.2 mM MgCl$_2$, 1× Protease Inhibitor Cocktail) for 10 min. Lysates were scraped into a Dounce homogenizer with 500 µl sucrose EDTA buffer (2 mM EDTA, 500 mM sucrose). Lysates were homogenized with 30 passes and transferred to a 1.5 ml Eppendorf tube. Lysates were centrifuged at 1000 $g$ for 10 min at 4°C to generate a post-nuclear supernatant (PNS). The PNS concentration was determined using a BCA assay, and 300 µg in 1 ml of H$_2$O was centrifuged at 100,000 $g$ for 30 min to separate the S100 and P100 fractions. Lysates, S100 fractions and P100 fractions were run on an SDS-PAGE gel and analyzed using western blot analysis.

## Acknowledgements

Seahorse analysis was performed by the WVU Mitochondria, Metabolism & Bioenergetics Working Group. The authors would like to thank Ethan Meadows, Madelyn Logan, and Bradley Lokant for their excellent technical assistance.

## Competing interests

B.A.W. was funded by a research grant from Genentech, Inc., a Roche Company. The authors declare no other competing or financial interests.

## Author contributions

Conceptualization: H.L.H., B.A.W.; Data curation: B.A.W.; Formal analysis: H.L.H., B.A.W.; Funding acquisition: B.A.W.; Investigation: H.L.H., B.A.W.; Methodology: H.L.H., B.A.W.; Project administration: B.A.W.; Supervision: B.A.W.; Validation: H.L.H., B.A.W.; Visualization: H.L.H., B.A.W.; Writing – original draft: H.L.H., B.A.W.; Writing – review & editing: H.L.H., B.A.W.

## Funding

This work was supported by West Virginia University start-up funding, Visual Sciences CoBRE project leader funding (P20GM144230), and National Institute of General Medical Sciences funding (R35GM158392) to B.A.W. The content is solely the responsibility of the authors and does not necessarily represent the official views of the National Institutes of Health. Open Access funding provided by West Virginia University. Deposited in PMC for immediate release.

## Data and resource availability

All relevant data and details of resources can be found within the article and its supplementary information.

## Peer review history

The peer review history is available online at https://journals.biologists.com/jcs/lookup/doi/10.1242/jcs.264251.reviewer-comments.pdf

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
