## [Peer Review File · Journal of Cell Science]

Functional requirements of the liver isoform of phosphofructokinase-1 in breast cancer cell migration

Heather L. Hansen and Bradley A. Webb

DOI: 10.1242/jcs.264251

Editor: Guillaume Jacquemet

Review timeline

Original submission:	27 June 2025
Editorial decision:	29 July 2025
First revision received:	22 September 2025
Accepted:	11 October 2025

Original submission

First decision letter

MS ID#: jcs.264251

MS TITLE: Functional requirements of the liver isoform of phosphofructokinase-1 in breast cancer cell migration

AUTHORS: Heather Hansen; Bradley Webb

ARTICLE TYPE: Research Article

Dear Dr Webb,

We have now reached a decision on the above manuscript.

To see the reviewers' reports and a copy of this decision letter, please go to:

Reviewer 1

This manuscript represents a significant insight into the role of localized glycolysis in fueling the high-energy demands of lamellipodial-based migration and chemotaxis. I found the paper to be quite complete in terms of the breadth of the data presented, effectively connecting localization of PFKL within migrating cells, PFKL activity and downstream changes to metabolism, and changes to cell behavior. My main concerns were centered on the interpretation of the presence of absence of PFKL and the PFKL constructs to lamellipodia, namely the lack of quantification of this localization in several instances, along with not testing whether the protein was no longer recruited to lamellipodia versus the formation of lamellipodia were compromised. By clarifying these issues, I think the authors will have made a very solid contribution to our fundamental understanding of single cell motility.

Suggestions:

I have a few suggestions on how to reinforce the conclusions that PFKL filament formation is required for PFKL recruitment to lamellipodia and chemotaxis towards EFG. While the images are clear and convincing, I would like the authors to provide at least a qualitative description of how many cells they saw similar colocalization in lamellipodia while reporting the total number of cells imaged and number of independent experimental replicates. I don't think the typical co-localization

analysis (Pearson's correlation coefficient) would be very effective here, since it is a relatively minor amount of the total protein enriched in lamellipodia. I did find the approach used in Fig. 5D to be quite effective. I suggest the authors incorporate an additional control experiment with FLAG-PFKL in serum starved cells, prior to the addition of EGF. They could then compare the cytoplasmic/leading edge ratios to demonstrate a clear enrichment of PFKL at the leading edge when lamellipodia are present following the addition of EGF.

In figure 2, please indicate number of independent replicates of siRNA knockdown quantification in the figure legend. Along with p-values.

The loss of lamellipodia enrichment of PFKL when its filament forming ability is compromised is quite intriguing. I suggest the authors take this a step further and determine whether it is the targeting of PFKL to lamellipodia that has been disrupted, or the formation of lamellipodia in general has been reduced. To accomplish this the author's could stain the FLAG-PFKL-NL02T expressing cells with a lamellipodial marker (such as cortactin, cofilin, or Arp2/3) and determine whether the number/size of lamellipodia have been disrupted as well. If lamellipodia formation were to be disrupted, it would be interesting to determine whether another F-actin structure, such as cortical actin near the nucleus or stress fibers were also disrupted. This type of information could indicate how global or specific the need for PFKL activity is in migrating cells.

I was interested in hearing the author's thoughts on why disruption of PFKL filament formation and lamellipodial enrichment abolished chemotaxis, but did not reduce overall velocity. Doesn't it suggest that the regulation of overall cell speed by PFKL occur somewhere other than lamellipodia, perhaps in another area of intense energy consumption during cell migration? But that energy usage in lamellipodia could be more immediately related to direction sensing? This is what occurred to me, but adding a small paragraph highlighting this issue in the discussion would enrich the paper.

I'm also curious if their timelapse movies show any differences in protrusion morphology or dynamics that could point the way towards future mechanistic studies of how PFKL activity is facilitating directional migration. I would leave this up to the discretion of the authors, but if disrupting PFKL activity or localization caused changes such as bigger lamellipodia, multiple protrusions, or slower lamellipodial extension-retraction dynamics, it could be worthwhile to report here.

This sentence from the first paragraph of the results appears to be missing a citation: "We chose MDA-MB-231 breast cancer cells, as they are highly migratory and are shown to rely heavily on upregulated glycolysis to sustain their increased cell proliferation and migration."

Shouldn't the sentence at the end of page 10 be written (swap the order of the assays to reflect the ratio reported in the figure: "However, the ratio of mitochondrial activity (oxygen consumption rate; OCR) to glycolytic activity (ECAR) was slightly elevated in siPFKL transfected cells.."

Reviewer 2

Summary

This manuscript by Hansen and Webb investigates the role of the glycolytic enzyme phosphofructokinase-1 (PFKL) in directional cell migration. The authors performed various loss-of-function studies (silencing, inhibitor treatment, overexpression of mutants) to show that PFKL catalytic activity is required for directional migration of MDA-MB-231 cells and that PFKL filamentation is important to localize PFKL in lamellipodia to support directional migration. The study aims to contribute to our understanding of the role of the glycolytic enzyme PFKL in breast cancer cell chemotaxis.

Overall Assessment

The topic is novel, interesting, timely and relevant to the general field of cell biology. The experimental approach is largely appropriate. However, I have several comments regarding quantifications, repetitions and statistics of experiments, as well as a general comment about the

role of glycolysis in the phenotype. Moreover, in the discussion the authors fail to connect the conclusions of all their findings. Those concerns should be addressed prior to publication.

Note: Page numbers are missing and the lines are incorrectly aligned making it difficult to precisely mention the comments and corrections. The page numbers below are based on the pdf file and the line numbers are approximate.

Major Comments

1. All immunofluorescent images and Western blots (WB) lack quantitative analysis. Including quantification with appropriate statistics would strengthen the conclusions. The number of independent experiments performed should always be mentioned, in addition to the number of replicates/images analyzed per experiment.

In the case that data are coming only from one experiment, then those experiments need to be repeated.

Images that need quantification, statistics or mention the N number: Figure 1, Figure 2A, Figure 2D & E, Figures 3A, 3D & E, Figure 4 B & C and Figure 5D, E, F & G.

In Figure 5E for example the authors mention in page 15 line 12 that there is "a significant decrease", a claim that cannot be supported without quantification.

2. The role of glycolysis in the phenotype should be explained and discussed.

In Supplem Fig 1B it is shown that the other PFK1 isoforms are not increased to compensate for the loss of PFKL - if anything, their expressions is even inhibited. These data are in agreement with Supplem Fig 1C data showing that total PFK1 activity is inhibited by 50%. However, inhibition of PFK1 by 50% does not have any effect on glycolysis as shown in Fig 2B, suppl Fig 1D. There is only a modest difference in OCR/ECAR (Fig 2C). Given that PFK1 is a rate-limiting, "gatekeeper" enzyme in glycolysis, the authors should explain these unexpected findings. This is particularly important also because the authors concluded in Figure 1 that glycolysis is compartmentalized to meet local energy demands during migration.

One explanation mentioned by the authors is possible compensation by other PFK1 isoforms (page 10, line 52), but this is not supported by their findings (suppl Fig 1B). So, they should remove this explanation.

Page 11 line 12 the authors mention "despite the minor impact on glycolytic flux". This should be rephrased as the glycolytic flux per se is not measured and based on the Seahorse experiment and lactate measurements there is no effect in glycolysis. In addition, it is shown in Suppl Fig 1F that proliferation is decreased, but an argumentation and possible explanation of this finding is lacking.

In Figure 3B PFKL overexpression (PFKL WT) increased glycolysis compared to control, whereas catalytically inactive mutant decreased glycolysis compared to EGFR control. These data show that altering the expression and catalytic activity of PFKL affects glycolysis, as expected. Based on these data, it is even more surprising that siPFKL has no impact on glycolysis.

3. The authors should discuss their view on the contribution of the catalytic function of PFKL and the localization of PFKL in chemotaxis. Disrupting the catalytic activity does not affect the localization of PFKL in the cytoplasm (Fig 3C) but inhibits directional migration by approximately 50% (Fig 3D). On the other hand, disrupting filament formation reduces the localization of PFKL in the lamellipodia (Fig 5C) and inhibited directional migration by approximately 40%. Do the authors think that PFKL activity and localization are required for directional migration and why?

4. Given that no immunofluorescence stainings of endogenous PFKL are provided, it is assumed that there are no specific antibodies for PFKL. Please mention that in the text. Nevertheless, it would be advised to provide immunofluorescence images of siCTRL and siPFKL cells stained with a PFK1 Ab (staining total PFK1). It has been shown previously that PFK1 is localized in lamellipodia and it would be interesting to see if this is the case in MDA-MB-231 cells and how is the localization affected by PFKL silencing.

5. In Figure 2A PFKL levels increase over time in control cells. Please provide an explanation for that.

6. In Figure 3A the expression of FLAG-PFKL WT is very high. It is mentioned in the text that it is 15x higher than the control, but it seems to be much higher than that. Proper quantification of the bands is needed. Is FLAG-PFKL WT (Fig 3a) and FLAG-PFKL (Suppl Fig 1A) the same construct? Please also explain whether these supra-physiological levels could have an effect on the cells.

7. In Fig 3D despite the big increase in PFKL level by PFKL WT there was no effect on migration directionality and that can be expected given that all the control cells move towards EGF and additional PFKL does not have an impact. However, given that siPFKL affected migration velocity the authors should comment why PFKL overexpression does not increase velocity.

Minor Comments

1. Supplementary Fig 1A should rather move to the main figures and Figure 4A should rather move to the supplements
2. In Figure 2A it should be not mentioned more clearly which are the conditions compared (siPFKL compared to siCTRL?)
3. It should be explained what does glycolysis, glycolytic capacity and glycolytic reserve refer to. A suggestion is to using the Seahorse ECAR graph eg Suppl Fig 1C to explain.
4. Figure 3A shows the WB of FLAG-PFKL WT, and FLAG-PFKL-H199Y expression. Please provide unmodified blot for anti-FLAG.

Reviewer 3

SUMMARY OF THE ADVANCE MADE IN THIS PAPER AND ITS POTENTIAL SIGNIFICANCE TO THE FIELD

This study explores the role of the liver isoform phosphofructokinase-1 in cancer cell migration. Cancer cells are highly motile, which is energetically demanding, and cells rely heavily on an increased glycolysis to generate ATP needed for the processes such as actin cytoskeleton remodelling, myosin contraction and focal adhesion turnover. Prior studies have demonstrated that glycolytic pathway enzymes localize to membrane areas and stimulate cell migration. Since PFKL was previously shown to localise to lamellipodia together with other glycolytic pathway enzymes, authors tested its contribution to directional cell migration. They reported that silencing of PFKL impairs both migration speed and chemotactic accuracy. Catalytic activity of PFKL was also crucial, as shown by the expression of catalytically inactive mutant which exerted a similar phenotype. Treatment with an inhibitor targeting the allosteric activator of PFK also lowered the migration velocity and chemotaxis index, but to a lesser extent. Authors also used the filament formation-incompetent mutant, which failed to localize to lamellipodia and remained predominantly cytosolic. This mutant exhibited a reduced chemotaxis index compared to wild-type, but the migration velocity was normal. The results that differentiate the roles of catalytic function and filament formation were particularly interesting, and would be beneficial to expand upon.

The manuscript is very well written, with clear and logical flow. The study is offering valuable insight for the field since the role of glycolytic pathway enzymes in chemotaxis and cell migration in general is still somewhat underexplored. However, I have a concern about the specificity of the observed phenotype with the current experimental setup, which relies on siRNA, isoform nonspecific inhibitor, and overexpression of recombinant proteins. At the moment, the data does not fully support all the conclusions drawn. The authors would be encouraged to revise or moderate some of the claims accordingly.

SUGGESTIONS TO AUTHORS

Major comments

Major points I would like to raise:

1. Supplement 1B - the antibody appears to predominantly recognize PFKM, and western blot suggests that PFKL silencing also partially reduces PFKM levels (and possibly also PFKP). How much of the observed phenotype can be attributed to knockdown of PFKM/PFKP? If the PFKL knock-out strain was generated, this would be avoided, and provide clearer interpretation of the data. Did the authors consider generating a knockout for PFKL after the initial siRNA experiments?

2. Generation of stable cell lines - description was brief, so please correct me if I am wrong, but it looks like the authors used wild-type cells with endogenous PFKL, and transduced them using a lentiviral vector. Given the random integration of the constructs, this approach can lead to unpredictable effects (dosage effect or variable expression levels of the recombinant gene), as seen in the results. In line with the previous point, if the PFKL knockout strain was generated, it would allow for the replacement of the endogenous gene with tagged recombinant PFKL variants, giving a more controlled and physiologically relevant expression. I believe that would strengthen the study; results would be specific for PFKL isoform and would avoid potential influence of overexpression and dominant negative effects.

To summarize, I think generating a PFKL knockout strain would be a more rigorous approach if authors want to keep the focus specifically on the PFKL isoform. The current data support the requirement of functional PFK for chemotaxis, but do not clearly differentiate between the influence of PFKL vs the other isoforms. However, I acknowledge that it can be time consuming and potentially not easily feasible. Alternatively, the language could be toned down to acknowledge these limitations more clearly.

Minor comments

1. The inhibition of glycolysis was shown to attenuate cell motility in previous research. Here, siPFKL had only minor impact on glycolytic flux but dramatic effect on chemotaxis, do the authors offer an explanation?

2. Dramatic decrease of cell proliferation in siPFKL samples is surprising, particularly given that proliferation was unaffected in strains expressing the PFKL mutants, for example. Could it be nonspecific effect?

3. Chemotaxis assays for siPFKL were started at 48h post-transfection to mitigate the impact on cell proliferation, but the assays themselves were done for 16h - during this time the impact on cell proliferation becomes more pronounced. Could this influence the results of chemotaxis assays? Were the cells viable and only dividing more slowly, or was there also cell death? Did the authors observe changes in cell viability, morphology or migratory behaviour over the course of the assay? Are there accompanying videos of chemotaxing cells that would show the cells during these experiments?

4. Fig 3c - although this is a single image, it appears that recruitment of PFKL to lamellipodia is enhanced when PFKL-H199Y is expressed, together with increased actin enrichment. Did you observe something similar in other cells or is this just a coincidence? Did the authors perform any quantification of PFKL localisation (lamellipodia vs cytoplasm) in these strains?

5. There is a typo in the sentence " To ask if filament formation alters the cellular distribution of PFKL in our present model, we generated MDA-MB-231 cell lines stably expressing filament incompetent FLAG-PFKL-N702T using lentiviral transduction and confirmed expressing using western blot analysis (Fig. 5A)."

6. Reference no 12 (Zhan, H.; Pal, D. S.; Borleis, J.; Janetopoulos, C.; Huang, C.-H.; Devreotes, P. N. Self-Organizing Glycolytic Waves Fuel Cell Migration and Cancer Progression. *bioRxiv* 2024, 2024.01.28.577603. <https://doi.org/10.1101/2024.01.28.577603>) has now been published

First revisionAuthor response to reviewers' comments**Reviewer 1:**

This manuscript represents a significant insight into the role of localized glycolysis in fueling the high-energy demands of lamellipodial-based migration and chemotaxis. I found the paper to be quite complete in terms of the breadth of the data presented, effectively connecting localization of PFKL within migrating cells, PFKL activity and downstream changes to metabolism, and changes to cell behavior. My main concerns were centered on the interpretation of the presence or absence of PFKL and the PFKL constructs to lamellipodia, namely the lack of quantification of this localization in several instances, along with not testing whether the protein was no longer recruited to lamellipodia versus the formation of lamellipodia were compromised. By clarifying these issues, I think the authors will have made a very solid contribution to our fundamental understanding of single cell motility.

We thank the reviewer for their thorough and thoughtful suggestions. We have added new data and modified our interpretation to address these concerns. Specifically:

- 1) Lack of quantification - we have added new data to quantify the enrichment of PFKL and mutant to the lamellipodia. These data are found on page 12 and Fig S3 D.
- 2) Interpretation of data - we have modified the discussion to better reflect our research question, results and conclusion. This is found on page 12-15
- 3) Recruitment to vs formation of lamellipodia -Based on our current results, we cannot speak to the effect of PFKL recruitment to the membrane on lamellipodia dynamics. We believe this is an important, however analysis of fixed cells will not be able to properly answer this question due to the highly dynamic nature of lamellipodia in MDA-MB-231 cells. Live-cell analysis of lamellipodia dynamics under serum starved and EGF stimulated conditions will be needed to properly answer this questions. We believe this lay outside the scope of the current paper. Instead, we have modified the text to acknowledge this caveat and tone down our interpretation to what is supported by our results.

Suggestions:

- I have a few suggestions on how to reinforce the conclusions that PFKL filament formation is required for PFKL recruitment to lamellipodia and chemotaxis towards EGF. While the images are clear and convincing, I would like the authors to provide at least a qualitative description of how many cells they saw similar colocalization in lamellipodia while reporting the total number of cells imaged and number of independent experimental replicates. I don't think the typical co-localization analysis (Pearson's correlation coefficient) would be very effective here, since it is a relatively minor amount of the total protein enriched in lamellipodia. I did find the approach used in Fig. 5D to be quite effective. I suggest the authors incorporate an additional control experiment with FLAG-PFKL in serum starved cells, prior to the addition of EGF. They could then compare the cytoplasmic/leading edge ratios to demonstrate a clear enrichment of PFKL at the leading edge when lamellipodia are present following the addition of EGF.

We thank the reviewer for highlight an issue with our scientific communication. We do not claim that EGF is required for the recruitment of PFKL to lamellipodia. Identifying the mechanism of recruitment is an important goal to understanding the function of the enzyme in cell migration. We are currently trying to determine how PFKL is recruited, with the goal of generating a mutant that cannot localize to the leading edge. However, this is a long-term project outside the scope of the current manuscript.

To better communicate our findings, we have reorganized the manuscript to highlight the role of PFKL in cell migration. This was achieved by moving the siRNA data and adding new shRNA data to Figure 1. This is now supported by fixed-cell imaging in Figure 2. We have added more data

quantifying the lamellipodia-to- cytoplasm ratio of PFKL in wild type and N702T PFKL expressing cells as well as quantifying the number of EFG-stimulated cells with PFKL wt and N702T localized to the lamellipodia. These data are found in Figure 5D and Supplemental Figure 3D.

- The loss of lamellipodia enrichment of PFKL when its filament forming ability is compromised is quite intriguing. I suggest the authors take this a step further and determine whether it is the targeting of PFKL to lamellipodia that has been disrupted, or the formation of lamellipodia in general has been reduced. To accomplish this the authors could stain the FLAG-PFKL-NL02T expressing cells with a lamellipodial marker (such as cortactin, cofilin, or Arp2/3) and determine whether the number/size of lamellipodia have been disrupted as well. If lamellipodia formation were to be disrupted, it would be interesting to determine whether another F-actin structure, such as cortical actin near the nucleus or stress fibers were also disrupted. This type of information could indicate how global or specific the need for PFKL activity is in migrating cells.

Lamellipodia in MDA-MB-231 cells are very dynamic and heterogenous, which makes this analysis difficult on fixed cells. Instead, live-cell imaging should be performed. Further, we are asking if the localization of PFKL to cells is unique to breast cancer cells or is a more generalized phenomenon. Our preliminary results suggest that PFKL is localized to cells of various origins in tissue culture. Use of different cell types, such as keratinocytes or epithelial cells that migrate with large, persistent lamellipodia to answer this question is an area of active research.

- I was interested in hearing the author's thoughts on why disruption of PFKL filament formation and lamellipodial enrichment abolished chemotaxis, but did not reduce overall velocity. Doesn't it suggest that the regulation of overall cell speed by PFKL occur somewhere other than lamellipodia, perhaps in another area of intense energy consumption during cell migration? But that energy usage in lamellipodia could be more immediately related to direction sensing? This is what occurred to me, but adding a small paragraph highlighting this issue in the discussion would enrich the paper.

Text has been added to the discussion section to explore this idea.

- I'm also curious if their timelapse movies show any differences in protrusion morphology or dynamics that could point the way towards future mechanistic studies of how PFKL activity is facilitating directional migration. I would leave this up to the discretion of the authors, but if disrupting PFKL activity or localization caused changes such as bigger lamellipodia, multiple protrusions, or slower lamellipodial extension-retraction dynamics, it could be worthwhile to report here.

Due to technical limitations, including low magnification images (20X) and the plastic substrate of the chemotaxis chambers, the image quality is not sufficient for reproducible and quantitative assessment of lamellipodial dynamics. Future live-cell experiments using higher magnification images and appropriate imaging substrate are planned to answer these questions.

- In figure 2, please indicate the number of independent replicates of siRNA knockdown quantification in the figure legend. Along with p-values.

The number of independent replicates was added to legend, and p-values are shown in the figure.

- This sentence from the first paragraph of the results appears to be missing a citation: "We chose MDA- MB-231 breast cancer cells, as they are highly migratory and are shown to rely heavily on upregulated glycolysis to sustain their increased cell proliferation and migration."

This has been corrected.

- Shouldn't the sentence at the end of page 10 be written (swap the order of the assays to reflect the ratio reported in the figure: "However, the ratio of mitochondrial activity (oxygen consumption rate; OCR) to glycolytic activity (ECAR) was slightly elevated in siPFKL transfected cells..."

This has been corrected.

Reviewer 2:

Summary

This manuscript by Hansen and Webb investigates the role of the glycolytic enzyme phosphofructokinase-1 (PFKL) in directional cell migration. The authors performed various loss-of-function studies (silencing, inhibitor treatment, overexpression of mutants) to show that PFKL catalytic activity is required for directional migration of MDA-MB-231 cells and that PFKL filamentation is important to localize PFKL in lamellipodia to support directional migration. The study aims to contribute to our understanding of the role of the glycolytic enzyme PFKL in breast cancer cell chemotaxis.

Overall Assessment

The topic is novel, interesting, timely and relevant to the general field of cell biology. The experimental approach is largely appropriate. However, I have several comments regarding quantifications, repetitions and statistics of experiments, as well as a general comment about the role of glycolysis in the phenotype.

Moreover, in the discussion the authors fail to connect the conclusions of all their findings. Those concerns should be addressed prior to publication.

Note: Page numbers are missing and the lines are incorrectly aligned making it difficult to precisely mention the comments and corrections. The page numbers below are based on the pdf file and the line numbers are approximate.

We thank the reviewer for their thorough review and appreciate the attention to statistical details. We agree that some statements may be over-interpreted, and these have been addressed in the manuscript.

Major Comments

1. All immunofluorescent images and Western blots (WB) lack quantitative analysis. Including quantification with appropriate statistics would strengthen the conclusions. The number of independent experiments performed should always be mentioned, in addition to the number of replicates/images analyzed per experiment.

In the case that data are coming only from one experiment, then those experiments need to be repeated. Images that need quantification, statistics or mention the N number: Figure 1, Figure 2A, Figure 2D & E, Figures 3A, 3D & E, Figure 4 B & C and Figure 5D, E, F & G.

Figure 1 (now 2): Quantification of cells with PFKL and PFKM or HK2 colocalization is now listed in the text (Page 10)

Figure 2A (Now 1A): Percent of PFKL depletion is listed in the figure with SEM and p-values. The number of biological replicates is listed in the figure legend.

Figures 2D (now 1D), 3D, 4B, and 5F (spider diagrams): Number of cells tracked, as well as number moving left vs right is shown in figure. The percentage of cells traveling left versus right is located in the results section. The total number of cells tracked is additionally listed in figure legend.

Figure 2E (Now 1E), 3E, 4C, 5G (chemotaxis parameter graphs): The number of biological replicates is listed in the figure legend. A key for p-values and type of statistical test run is in the figure legend. Details about analysis of the chemotaxis assays are also found in the methods section.

Figure 3A: (mean \pm SEM) of exogenous PFKL expression has been added to Figure 1A

Figure 3C: Quantification of lamellipodia: cytoplasm intensity ratio has been done and has been added as supplemental figure 2D

Figure 5D: The total number of cells, number of biological replicates, p-value, and statistical test

performed is described in the figure legend.

Figure 5E: The number of replicates, p-value key, and type of statistical test performed is described in the figure legend

We have also ensured that we are using the proper statistical tests for the chemotaxis plots (velocity, X-FMI, and Y-FMI) according to manufacturer recommendation (PMID: 21592329, PMID: 30212492).

In Figure 5E for example the authors mention in page 15 line 12 that there is "a significant decrease", a claim that cannot be supported without quantification.

Quantification of presence of wild type and N702T FLAG-PFKL in the soluble (S100) and crude particulate (P100) fractions is found in Figure 5E. Densitometry was performed using ImageJ, and significance was determined using paired t-test, as described in the methods section and figure legend.

2. The role of glycolysis in the phenotype should be explained and discussed.

In Supplem Fig 1B it is shown that the other PFK1 isoforms are not increased to compensate for the loss of PFKL - if anything, their expression is even inhibited. These data are in agreement with Supplem Fig 1C data showing that total PFK1 activity is inhibited by 50%. However, inhibition of PFK1 by 50% does not have any effect on glycolysis as shown in Fig 2B, suppl Fig 1D. There is only a modest difference in OCR/ECAR (Fig 2C). Given that PFK1 is a rate-limiting, "gatekeeper" enzyme in glycolysis, the authors should explain these unexpected findings. This is particularly important also because the authors concluded in Figure 1 that glycolysis is compartmentalized to meet local energy demands during migration. One explanation mentioned by the authors is possible compensation by other PFK1 isoforms (page 10, line 52), but this is not supported by their findings (suppl Fig 1B). So, they should remove this explanation.

PFK1 is allosterically activated by signals of low cellular energy, such as AMP and ADP. So even though there is a decrease in protein levels, the cell may be able to compensate by allosterically activating the remaining PFK1. It is possible that other metabolic pathways can contribute to ECAR, such as the production of CO₂ in the TCA cycle. This has been added to the text on page 8.

Page 11 line 12 the authors mention "despite the minor impact on glycolytic flux". This should be rephrased as the glycolytic flux per se is not measured and based on the Seahorse experiment and lactate measurements there is no effect in glycolysis.

This has been clarified in the text (page 8).

In addition, it is shown in Suppl Fig 1F that proliferation is decreased, but an argumentation and possible explanation of this finding is lacking. In Figure 3B PFKL overexpression (PFKL WT) increased glycolysis compared to control, whereas catalytically inactive mutant decreased glycolysis compared to EGFR control. These data show that altering the expression and catalytic activity of PFKL affects glycolysis, as expected. Based on these data, it is even more surprising that siPFKL has no impact on glycolysis.

This has been addressed in the text.

3. The authors should discuss their view on the contribution of the catalytic function of PFKL and the localization of PFKL in chemotaxis. Disrupting the catalytic activity does not affect the localization of PFKL in the cytoplasm (Fig 3C) but inhibits directional migration by approximately 50% (Fig 3D). On the other hand, disrupting filament formation reduces the localization of PFKL in the lamellipodia (Fig 5C) and inhibits directional migration by approximately 40%. Do the authors think that PFKL activity and localization are required for directional migration and why?

We thank the reviewer for bringing this point up, as it provided us with an opportunity to improve our science communication. We have clarified our conclusions in the discussion section to outline our proposed model in a more clear manner

We also clarified what the percentage of cells moving towards or away from the chemoattractant means. In these chemotaxis assays, 50% of cells moving towards the gradient indicates random migration, 100% of cells moving towards the gradient indicates full directional migration. In the examples given above, approximately 50% of PFKL-H199Y cells moving towards the gradient indicates that they are undergoing random migration. This is supported by the X-forward migration index of approximately -0.05 which is near zero (which indicates weak directional migration). Similarly, approximately 60% of PFKL-N702T cells moved towards the gradient as opposed to 40% away. Compared to the PFKL-WT expressing cells (89.5% towards, 10.5% away), these cells are migrating much more randomly. Clarifying statements for interpreting these result has been added to pages 9, 11, 12, and 13

4. Given that no immunofluorescence stainings of endogenous PFKL are provided, it is assumed that there are no specific antibodies for PFKL. Please mention that in the text. Nevertheless, it would be advised to provide immunofluorescence images of siCTRL and siPFKL cells stained with a PFK1 Ab (staining total PFK1). It has been shown previously that PFK1 is localized in lamellipodia and it would be interesting to see if this is the case in MDA-MB-231 cells and how is the localization affected by PFKL silencing.

In our hands, no available antibodies recognize all three isoforms with equal affinity. Commercial antibodies marketed as pan-specific are typically raised against PFKM and show variable affinity for PFKL and PFKP, which is unsurprising given that the three isoforms share only 70% amino acid identity. We have extensively tested multiple PFKL antibodies by western blot analysis against purified recombinant protein and consistently observe cross-reactivity among the three human isoforms.

In response to the reviewers concern, we purchased a new commercial PFKL antibody and attempted immunofluorescence staining in two systems: a PFKL CRISPR knockout HepG2 cell line and the shRNA- mediated PFKL-depleted MDA-MB-231 cell line described here. In both cases, we observed a strong, non- specific perinuclear staining. Furthermore, despite shPFKL cells retaining only ~10% of endogenous PFKL, they exhibited staining levels comparable to parental and shScramble controls. These results make it difficult to draw definitive conclusions, and for this reason, we have not included these data in our manuscript.

Two cells of each experimental group are pictured below.

5. In Figure 2A PFKL levels increase over time in control cells. Please provide an explanation for that.

The level of PFKL protein is regulated both transcriptionally and post-transcriptionally. Recent research suggests that substrate stiffness can influence PFKL abundance by sequestering a ubiquitin E3 ligase to stress fiber under conditions of higher substrate rigidity (PMID: 32051585). As our cells show increased confluency during the time course, we find an increase in PFKL abundance. A statement reflecting this result has been added to the results (page 8).

6. In Figure 3A the expression of FLAG-PFKL WT is very high. It is mentioned in the text that it is 15x higher than the control, but it seems to be much higher than that. Proper quantification of the bands is needed.

These bands were quantified by densitometry using ImageJ. A graph quantifying PFKL expression relative to loading control has been added to Fig. 3A

Is FLAG-PFKL WT (Fig 3a) and FLAG-PFKL (Suppl Fig 1A) the same

construct?

Yes, we have ensured that all figure labels are consistent throughout the manuscript.

Please also explain whether these supra-physiological levels could have an effect on the cells.

This is addressed in the results section (Page 11)

7. In Fig 3D despite the big increase in PFKL level by PFKL WT there was no effect on migration directionality and that can be expected given that all the control cells move towards EGF and additional PFKL does not have an impact. However, given that siPFKL affected migration velocity the authors should comment why PFKL overexpression does not increase velocity.

This has been clarified in the text (Page 11)

Minor Comments

1. Supplementary Fig 1A should rather move to the main figures and Figure 4A should rather move to the supplements

We thank the reviewer for the suggestion but prefer to keep the panels as they are in the current manuscript.

2. In Figure 2A it should be not mentioned more clearly which are the conditions compared (siPFKL compared to siCTRL?)

This has been updated in the figure legend.

3. It should be explained what glycolysis, glycolytic capacity and glycolytic reserve refer to. A suggestion is to use the Seahorse ECAR graph eg Suppl Fig 1C to explain.

This has been updated in Supp Fig 1D and the corresponding figure legend.

4. Figure 3A shows the WB of FLAG-PFKL WT, and FLAG-PFKL-H199Y expression. Please provide unmodified blot for anti-FLAG.

All unmodified blots are provided to the journal as per submission policy. The brightness and contrast was adjusted on anti-FLAG blot to visualize FLAG-H199Y. While this global adjustment was made for display purposes, analysis was performed on unmodified images. All image adjustments and analyses were performed within the JCS guidelines.

Reviewer 3: SUMMARY OF THE ADVANCE MADE IN THIS PAPER AND ITS POTENTIAL SIGNIFICANCE TO THE FIELD

This study explores the role of the liver isoform phosphofructokinase-1 in cancer cell migration. Cancer cells are highly motile, which is energetically demanding, and cells rely heavily on an increased glycolysis to generate ATP needed for the processes such as actin cytoskeleton remodelling, myosin contraction and focal adhesion turnover. Prior studies have demonstrated that glycolytic pathway enzymes localize to membrane areas and stimulate cell migration. Since PFKL was previously shown to localise to lamellipodia together with other glycolytic pathway enzymes, authors tested its contribution to directional cell migration. They reported that silencing of PFKL impairs both migration speed and chemotactic accuracy. Catalytic activity of PFKL was also crucial, as shown by the expression of catalytically inactive mutant which exerted a similar phenotype. Treatment with an inhibitor targeting the allosteric activator of PFK also lowered the migration velocity and chemotaxis index, but to a lesser extent. Authors also used the filament formation-incompetent mutant, which failed to localize to lamellipodia and remained predominantly

cytosolic. This mutant exhibited a reduced chemotaxis index compared to wild-type, but the migration velocity was normal. The results that differentiate the roles of catalytic function and filament formation were particularly interesting, and would be beneficial to expand upon. The manuscript is very well written, with clear and logical flow. The study is offering valuable insight for the field since the role of glycolytic pathway enzymes in chemotaxis and cell migration in general is still somewhat underexplored. However, I have a concern about the specificity of the observed phenotype with the current experimental setup, which relies on siRNA, isoform nonspecific inhibitor, and overexpression of recombinant proteins. At the moment, the data does not fully support all the conclusions drawn. The authors would be encouraged to revise or moderate some of the claims accordingly.

SUGGESTIONS TO AUTHORS

Major comments

Major points I would like to raise:

1. Supplement 1B - the antibody appears to predominantly recognize PFKM, and western blot suggests that PFKL silencing also partially reduces PFKM levels (and possibly also PFKP). How much of the observed phenotype can be attributed to knockdown of PFKM/PFKP? If the PFKL knock-out strain was generated, this would be avoided, and provide clearer interpretation of the data. Did the authors consider generating a knockout for PFKL after the initial siRNA experiments?
2. Generation of stable cell lines - description was brief, so please correct me if I am wrong, but it looks like the authors used wild-type cells with endogenous PFKL, and transduced them using a lentiviral vector. Given the random integration of the constructs, this approach can lead to unpredictable effects (dosage effect or variable expression levels of the recombinant gene), as seen in the results. In line with the previous point, if the PFKL knockout strain was generated, it would allow for the replacement of the endogenous gene with tagged recombinant PFKL variants, giving a more controlled and physiologically relevant expression. I believe that would strengthen the study; results would be specific for PFKL isoform and would avoid potential influence of overexpression and dominant negative effects.

To summarize, I think generating a PFKL knockout strain would be a more rigorous approach if authors want to keep the focus specifically on the PFKL isoform. The current data support the requirement of functional PFK for chemotaxis, but do not clearly differentiate between the influence of PFKL vs the other isoforms. However, I acknowledge that it can be time consuming and potentially not easily feasible. Alternatively, the language could be toned down to acknowledge these limitations more clearly.

We thank the reviewer for their thorough and thoughtful suggestions. We have added new data and modified our interpretation to address these concerns. Specifically:

1 - We have generated a PFKL knockdown strain using shRNA. We show that these cells have approximately 10% of PFKL expression compared to parental cells, with no increase in PFKM or PFKP levels observed (Fig S1F and G). Importantly, the migration phenotype of shPFKL cells mirrored those of siPFKL (Fig. 1D-G).

These results support our hypothesis that this is a PFKL-specific effect. However, we have also toned down our language about isoform specificity.

2 - We have added a brief description of generation of stable FLAG-PFKL cell lines in the materials and methods (page 3). Cells were transduced with lentivirus expressing FLAG-PFKL underwent antibiotic selection. A pool of antibiotic-resistant cells were used for cell experiments. As such, cells will express the endogenous protein. Although we have not tested for the integration site, it is unlikely that random integration of the vector occurred in the endogenous locus in both alleles in the pool of cells.

We agree that long-term experiments using cas9-mediated genetic editing to generate knockout and point mutants is an important next-step. However, we believe these experiments

lie outside the scope of the current manuscript. We are encouraged by the fact that we have such a drastic phenotype upon expression of FLAG- PFKL-H199Y or FLAG-PFKL-N702T, and hypothesize that performing these experiments in a clean background would give cleaner results.

Minor comments

1. The inhibition of glycolysis was shown to attenuate cell motility in previous research. Here, siPFKL had only minor impact on glycolytic flux but dramatic effect on chemotaxis, do the authors offer an explanation?

This has been clarified in the results section (Page 8).

2. Dramatic decrease of cell proliferation in siPFKL samples is surprising, particularly given that proliferation was unaffected in strains expressing the PFKL mutants, for example. Could it be nonspecific effect?

To address these concerns, we have generated a stable lentiviral shRNA PFKL knockdown cell line. In contrast to siRNA knockdown assays, no significant difference in proliferation rate was observed in shPFKL cells when compared to parental or control shScramble transduced cell lines. The cause of this discrepancy is not known but may be caused by off-target effects of the PFKL siRNA or long-term adaptation of the cell lines and is an area of future inquiry. Determining the cause of this discrepancy is an area of active research. We have added a statement to the results conveying these possibilities (Page 9).

3. Chemotaxis assays for siPFKL were started at 48h post-transfection to mitigate the impact on cell proliferation, but the assays themselves were done for 16h - during this time the impact on cell proliferation becomes more pronounced. Could this influence the results of chemotaxis assays? Were the cells viable and only dividing more slowly, or was there also cell death? Did the authors observe changes in cell viability, morphology or migratory behaviour over the course of the assay? Are there accompanying videos of chemotaxing cells that would show the cells during these experiments?

To address this important point, we generated a shRNA PFKL knockdown cell line that has no significant difference in proliferation when compared to non-transfected or control shScrambled transduced cells. Cells with shRNA depletion of PFKL showed the same chemotaxis properties observed in the PFKL siRNA transfected cells. Specifically, a significant decrease in velocity of migration and a loss of directional sensing. From these experiments, we conclude that differences in proliferation and viability did not impact the migration parameter measured in our assays.

Additionally, siRNA knockdown chemotaxis assays were performed at 48 hours to allow for significant depletion of PFKL. As this is a single cell chemotaxis assay, we were able to track migrating cells individually and avoid proliferating cells. We did not quantitatively assess cell viability in the chemotaxis chambers, however we noted that during our proliferation assays there was no difference in viability between the treatments.

4. Fig 3c - although this is a single image, it appears that recruitment of PFKL to lamellipodia is enhanced when PFKL-H199Y is expressed, together with increased actin enrichment. Did you observe something similar in other cells or is this just a coincidence?

The recruitment and function of PFKL in lamellipodia is an important question. One explanation is that PFKL can interact with F-actin, either directly or indirectly. The ability of PFK1 to interact with F-actin has been shown previously, with most studies focusing on the muscle isoform (PFKM) and more recently the platelet isoform (PFKP) (eg. PMIDs: 6446931, 35585241). It is unclear if PFKL directly interacts with F-actin filaments. On one hand, neither our lab nor our collaborators have observed a direct interaction with ATP- or ADP-bound F-actin in sedimentation assays of purified protein. On the other hand, PFKL cross-links to F-actin (PMID: 37806136) and our preliminary mass spectroscopy data shows that PFKL immunoprecipitated with F-actin and actin associated proteins. Whether PFKL directly or indirectly interacts with F-

actin is an area of active research by multiple labs. At this time, we do not have a conclusive answer to your question and, in the absence of further proof, consider the colocalization as coincidental.

Did the authors perform any quantification of PFKL localisation (lamellipodia vs cytoplasm) in these strains?

The quantification has been performed, and data is found in supplemental Figure 2D.

5. There is a typo in the sentence " To ask if filament formation alters the cellular distribution of PFKL in our present model, we generated MDA-MB-231 cell lines stably expressing filament incompetent FLAG-PFKL- N702T using lentiviral transduction and confirmed expressing using western blot analysis (Fig. 5A)."

This change has been made.

6. Reference no 12 (Zhan, H.; Pal, D. S.; Borleis, J.; Janetopoulos, C.; Huang, C.-H.; Devreotes, P. N. Self- Organizing Glycolytic Waves Fuel Cell Migration and Cancer Progression. bioRxiv 2024, 2024.01.28.577603. <https://doi.org/10.1101/2024.01.28.577603>) has now been published

This reference has been updated.

Second decision letter

MS ID#: jcs.264251R1

MS Title: Functional requirements of the liver isoform of phosphofructokinase-1 in breast cancer cell migration

Authors: Heather Hansen; Bradley Webb

Article Type: Research Article

Dear Dr Webb,

I am happy to tell you that your manuscript has been accepted for publication in Journal of Cell Science, pending standard publication integrity checks.